# Learning predictive cognitive maps with spiking neurons during behavior and replays

**Jacopo Bono†, Sara Zannone†, Victor Pedrosa, Claudia Clopath\***

Department of Bioengineering, Imperial College London, London, United Kingdom

**Abstract** The hippocampus has been proposed to encode environments using a representation that contains predictive information about likely future states, called the successor representation. However, it is not clear how such a representation could be learned in the hippocampal circuit. Here, we propose a plasticity rule that can learn this predictive map of the environment using a spiking neural network. We connect this biologically plausible plasticity rule to reinforcement learning, mathematically and numerically showing that it implements the TD-lambda algorithm. By spanning these different levels, we show how our framework naturally encompasses behavioral activity and replays, smoothly moving from rate to temporal coding, and allows learning over behavioral timescales with a plasticity rule acting on a timescale of milliseconds. We discuss how biological parameters such as dwelling times at states, neuronal firing rates and neuromodulation relate to the delay discounting parameter of the TD algorithm, and how they influence the learned representation. We also find that, in agreement with psychological studies and contrary to reinforcement learning theory, the discount factor decreases hyperbolically with time. Finally, our framework suggests a role for replays, in both aiding learning in novel environments and finding shortcut trajectories that were not experienced during behavior, in agreement with experimental data.

**\*For correspondence:**
c.clopath@imperial.ac.uk

†These authors contributed equally to this work

**Competing interest:** The authors declare that no competing interests exist.

## Editor's evaluation

This is an important article that leverages a spiking network model of the hippocampal circuit to show how spike-time-dependent plasticity can implement predictive reinforcement learning and form a predictive map of the environment. The authors provide a convincing and solid framework for understanding the prediction based learning rules that may be employed by the hippocampus to optimize an animal's behavior. This paper will be of interest to theoretical and experimental neuroscientists working on learning and memory as it provides new ways to connect computational models to experimental data that has yet to be fully explored from a reinforcement learning perspective.

## Introduction

Mid twentieth century, Tolman proposed the concept of cognitive maps (*Tolman, 1948*). These maps are abstract mental models of an environment which are helpful when learning tasks and in decision making. Since the discovery of hippocampal place cells, cells that are activated only in specific locations of an environment, it is believed that the hippocampus can provide the substrate to encode such cognitive maps (*O'Keefe and Dostrovsky, 1971*; *O'Keefe and Nadel, 1978*). More evidence of the role of the hippocampus in behavior was found in numerous experimental studies, such as the seminal water maze experiments (*Morris, 1981*; *Morris et al., 1982*), radial arm maze experiments (*Olton*

*and Papas, 1979*) as well as evidence of broader information processing beyond just cognitive maps (*Wood et al., 1999*; *Eichenbaum et al., 1999*; *Aggleton and Brown, 1999*; *Wood et al., 2000*).

While these place cells offer striking evidence in favour of cognitive maps, it is not clear what representation is actually learned by the hippocampus and how this information is exploited when solving and learning tasks. Recently, it was proposed that the hippocampus computes a cognitive map containing predictive information, called the successor representation (SR). Theoretically, this SR framework has some computational advantages, such as efficient learning, simple computation of the values of states, fast relearning when the rewards change and flexible decision making (*Dayan, 1993*; *Stachenfeld et al., 2014*; *Stachenfeld et al., 2017*; *Russek et al., 2017*; *Momennejad et al., 2017*). Furthermore, the SR is in agreement with experimental observations. Firstly, the firing fields of hippocampal place cells are affected by the strategy used by the animal to navigate the environment (known as the *policy* in machine learning), as well as by changes in the environment (*Mehta et al., 2000*; *Stachenfeld et al., 2017*). Secondly, reward revaluation — the ability to recompute the values of the states when rewards change — would be more effective than transition revaluation (*Russek et al., 2017*; *Momennejad et al., 2017*).

In this work, we study how this predictive representation can be learned in the hippocampus with spike-timing dependent synaptic plasticity (STDP). Using STDP at the mechanistic level, we show that the learning is equivalent to TD($\lambda$) on an algorithmic level. The latter is a well-studied and powerful algorithm known from reinforcement learning (*Sutton and Barto, 1998*), which we will discuss in more detail below.

Our model can thus learn over a behavioral timescale while using STDP timescales in the millisecond range. We show mathematically that our proposed framework smoothly connects a temporally precise spiking code akin to replay activity with a rate based code akin to behavioral spiking. Subsequently, we show that the delay-discounting parameter $\gamma$ allows us to consider time as a continuous variable, therefore we don't need to discretize time as is usual in reinforcement learning (*Doya, 1995*; *Doya, 2000*). Moreover, the delay-discounting in our model depends hyperbolically on time but exponentially on state transitions. We show how the $\gamma$ parameter can be modulated by neuronal firing rates and neuromodulation, allowing state-dependent discounting and in turn enabling richer information in the SR, such as the encoding of salient states, landmarks, reward locations, etc. Finally, replays have long been speculated to be involved in learning models of the environment, supported by experiments (*Johnson and Redish, 2007*; *Pfeiffer and Foster, 2013*; *Kay et al., 2020*) and models (*Hasselmo and Eichenbaum, 2005*; *Erdem and Hasselmo, 2012*; *Kubie and Fenton, 2012*). Here, we investigate how replays could play an additional role in learning the SR cognitive map. Following properties of TD($\lambda$), we show how we can achieve both low bias and low variance by using replays, translating to both quicker initial learning and convergence to lower error. We show how we can use replays to learn *offline*. In this way, policies can be refined without the need for actual exploration.

Our framework allows us to make predictions about the roles of behavioral learning and replay-like activity and how they can be exploited in representation learning. Furthermore, we uncover a relation between STDP and a higher level learning algorithm. Our work therefore spans the three levels of analysis proposed by *Marr, 2010*. On the implementational level, our model consists of a feedforward network of excitatory neurons with biologically plausible spike-timing dependent plasticity. On the algorithmic level, we show that our model learns the successor representation using the TD($\lambda$) algorithm. On the computational theory level, our model tackles representation learning using cognitive maps.

## Results

Cognitive maps are internal models of an environment which help animals to learn, plan and make decisions during task completion. The hippocampus has long been thought to provide the substrate for learning such cognitive maps (*O'Keefe and Dostrovsky, 1971*; *O'Keefe and Nadel, 1978*; *Morris, 1981*; *Morris et al., 1982*; *Wood et al., 1999*; *Eichenbaum et al., 1999*), and recent evidence points towards a specific type of representation learned by the hippocampus, the successor representation (SR) (*Stachenfeld et al., 2017*).

## The successor representation

In this section, we will give an overview of the successor representation and its properties, especially geared toward neuroscientists. Readers already familiar with this representation may safely move to the next section.

To understand the concept of successor representation (SR), we can consider a spatial environment — such as a maze — while an animal explores this environment. In this setting, the SR can be understood as how likely it is for the animal to visit a future location starting from its current position. We further assume the maze to be formed out of a discrete number of states. Then, the SR can be more formally described by a matrix with dimension ($N_{states} \times N_{states}$), where $N_{states}$ denotes the number of states in the environment and each entry $R_{ij}$ of this matrix describes the expected future occupancy of a state $S_j$ when the current state is $S_i$. In other words, starting from $S_i$, the more likely it is for the animal to reach the location associated with state $S_j$ and the nearer in the future, the higher the value of $R_{ij}$.

As a first example, we consider an animal running through a linear track. We assume the animal runs at a constant speed and always travels in the same direction — left to right (*Figure 1a*). We also split the track into four sections or *states*, $S_1$ to $S_4$, and the SR will be represented by a matrix with dimension ($4 \times 4$). Since the animal always runs from left to right, there is zero probability of finding the animal at position $i$ if its current position is greater than $i$. Therefore, the lower triangle of the successor matrix is equal to zero (*Figure 1b*). Alternatively, if the animal is currently at position $S_1$, it will be subsequently found at positions $S_2$, $S_3$, and $S_4$ with probability 1. The further away from $S_1$, the longer it will take the animal to reach that other position. In terms of the successor matrix, we apply a discounting factor $\gamma$ ($0 < \gamma \leq 1$) for each extra 'step' required by the animal to reach a respective location (*Figure 1b*).

Even though we introduced the linear track as an illustrative example, the SR can be learned in any environment (see *Figure 1—figure supplement 1* for an example in an open field). Note that the representation learned by the SR is dependent not only on the structure of the environment, but also on the policy — or strategy — used by the animal to explore the environment. This is because the successor representation is not purely concerned with the physical distance between two areas in the environment, but rather it measures how long it usually takes to reach one place when starting from the other. In this first example, the animal applied a deterministic policy (always running from left to right), but the SR can also be learned for stochastic policies. Furthermore, the SR is a multi-step representation, in the sense that it stores predictive information of multiple steps ahead.

Because of this predictive information, the SR allows sample-efficient re-learning when the reward location is changed (*Gershman, 2018*). In reinforcement learning, we tend to distinguish between model-free and model-based algorithms. The SR is believed to sit in-between these two modalities. In model-free reinforcement learning, the aim is to directly learn the value of each state in the environment. Since there is no model of the environment at all, if the location of a reward is changed, the agent will have to first unlearn the previous reward location by visiting it enough times, and only then it will be able to re-learn the new location. In model-based reinforcement learning, a precise model of the environment is learned, specifically, single-step transition probabilities between all states of the environment. Model-based learning is computationally expensive, but allows a certain flexibility. If the reward changes location it is immediate to derive the updated values of the states. As we have seen, however, the SR can re-learn a new reward location somewhat efficiently, although less so than model-based learning. The SR can also be efficiently learned using model-free methods and allows us to easily compute values for each state, which in turn can guide the policy (*Dayan, 1993*; *Russek et al., 2017*; *Momennejad et al., 2017*). This position between model-based and model-free methods makes the SR framework very powerful, and its similarities with hippocampal neuronal dynamics have led to increased attention from the neuroscience community. Finally, in our examples above we considered an environment made up of a discrete number of states. This framework can be generalised to a continuous environment represented by a discrete number of place cells.

## Learning the successor representation in biologically plausible networks

We propose a model of the hippocampus that is able to learn the successor representation. We consider a feedforward network comprising of two layers. Similar to *McNaughton and Morris, 1987*;

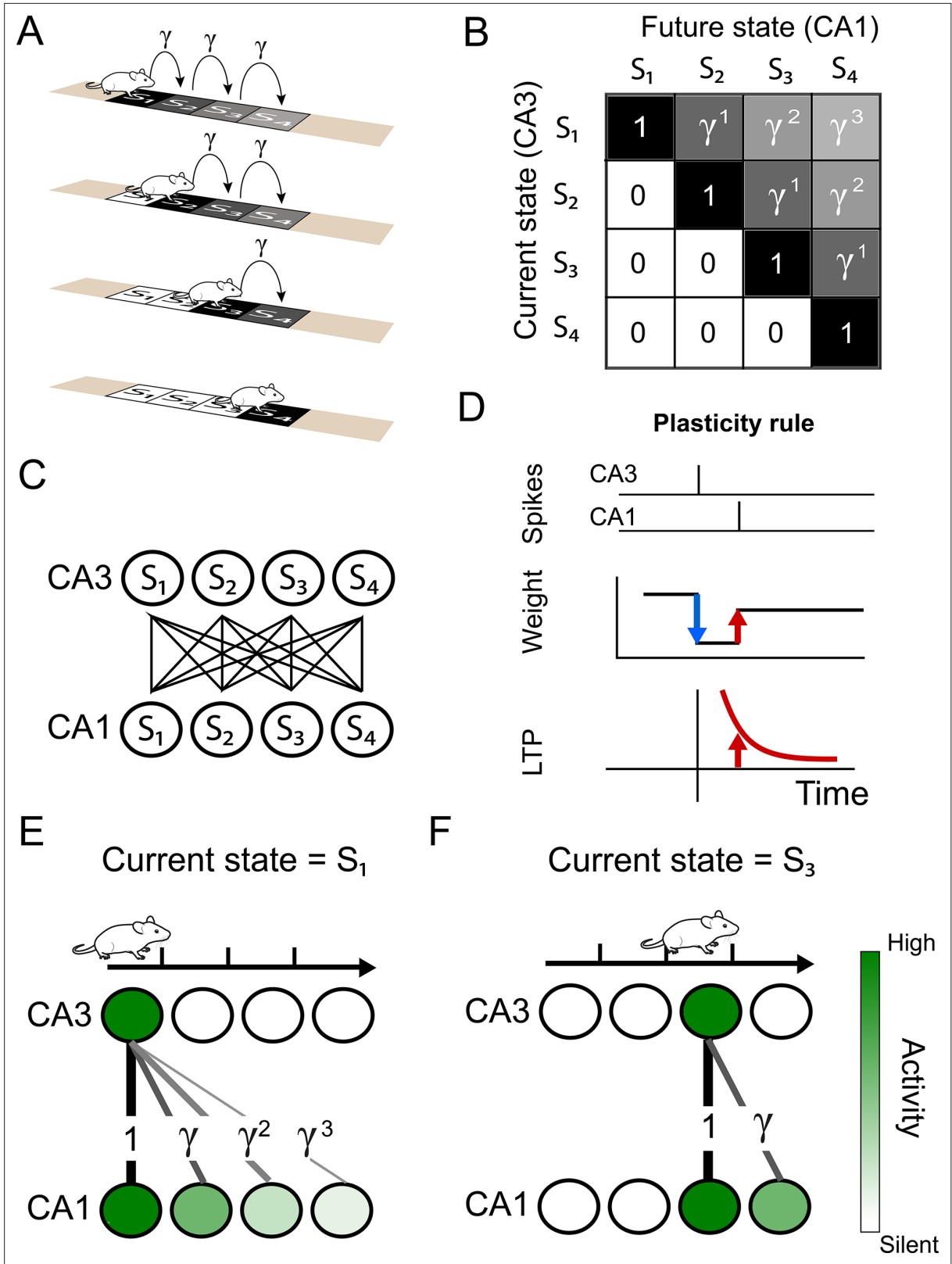

**Figure 1.** Successor representation and neuronal network. (**A**) Our simple example environment consists of a linear track with 4 states ($S_1$ to $S_4$) and the animal always moves from left to right — i.e. one epoch consists of starting in $S_1$ and ending in $S_4$. (**B**) The successor matrix corresponding to the task described in panel A. (**C**) Our neuronal network consists of a two layers with all-to-all feedforward connections. The presynaptic layer mimics hippocampal CA3 and the postsynaptic layer mimics CA1. (**D**) The synaptic plasticity rule consists of a depression term and a potentiation term. The

*Figure 1 continued on next page*

*Figure 1 continued*

depression term is dependent on the synaptic weight and presynaptic spikes (blue). The potentiation term depends on the timing between a pre- and post-synaptic spike pair (red), following an exponentially decaying plasticity window (bottom). (**E–F**) Schematics illustrating some of the results of our model. (**E**) Our spiking model learns the top row of the successor representation (panel B) in the weights between the first CA3 place cell and the CA1 cells. (**F**) Our spiking model learns the third row of successor representation (panel B) in the weights between the third CA3 place cell and the CA1 cells.

The online version of this article includes the following figure supplement(s) for figure 1:

**Figure supplement 1.** Learning the SR in a two-dimensional environment.

**Figure supplement 2.** The equivalence with TD($\lambda$) guarantees convergence even with random initial synaptic weights.

*Hasselmo and Schnell, 1994*; *Mehta et al., 2000*; *Hasselmo et al., 2002*, we assume that the presynaptic layer represents the hippocampal CA3 region and is all-to-all connected to a postsynaptic layer - representing the CA1 network (*Figure 1c*). The synaptic connections from CA3 to CA1 are plastic such that the weight changes follow a spike-timing-dependent plasticity (STDP) rule consisting of two terms: a weight-dependent depression term for presynaptic spikes and a potentiation term for pre-post spike pairs (*Figure 1d*).

For simplicity, we assume that the animal spends a fixed time $T$ in each state. During this time, a constant activation current is delivered to the CA3 neuron encoding the current location and, after a delay, to the corresponding CA1 place cell (see Materials and methods). On top of these fixed and location-dependent activations, the CA3 neurons can activate neurons in CA1 through the synaptic connections. In other words, the CA3 neurons are activated according to the current location of the animal, while the CA1 neurons have a similar location-dependent activity combined with activity caused by presynaptic neurons. The constant currents delivered directly to CA3 and CA1 neurons can be thought of as location-dependent currents from entorhinal cortex. These activations subsequently trigger plasticity at the synapses, and we can show analytically that, using the spike-timing dependent plasticity rule discussed above, the SR is learned in the synaptic weights (*Figure 1e and f*, and see Appendix).

Moreover, we find that, on an algorithmic level, our weight updates are equivalent to a learning algorithm known as TD($\lambda$), a powerful and well-known algorithm in reinforcement learning that can be used to learn the successor representation. TD($\lambda$) is based on a mixed methodology, which is regulated by the parameter $\lambda$. At one extreme, when $\lambda = 1$, the SR is estimated by taking the average of state occupancies over past trajectories. This type of algorithm is called TD(1) or Monte Carlo (MC). At the other extreme, when $\lambda = 0$, the estimate of the SR is adjusted 'online', with every step of the trajectory, by comparing the observed position with its predicted value. This algorithm is equivalent to TD(0). For all values of $\lambda$ in between, the algorithm employs a mixture of both methodologies. The extreme cases of TD(1) and TD(0) have different strengths and weaknesses, as we will discuss in more detail in the next sections.

In practice, we prove analytically the mathematical equivalence of the dynamics of our spiking neural network, and the TD($\lambda$) algorithm (see Appendix). Our calculations essentially prove that, at each step, our neural network tracks the reinforcement learning algorithm, known to converge to the theoretical values of the SR. This equivalence guarantees that our neural network weights will eventually converge to the correct SR matrix. As a proof of principle, we show that it is possible to learn the SR for any initial weights (*Figure 1—figure supplement 2*), independently of any previous learning in the CA3 to CA1 connections.

Importantly, from our analytical derivations (see Appendix), we find that the $\lambda$ parameter depends on the behavioral parameter T (the time an animal spends in a state). We find that, the larger the time T, the smaller the value of $\lambda$ and vice-versa. In other words, when the animal moves through the trajectory on behavioral time-scales (large T compared to the synaptic plasticity time-scales $\tau_{\mathrm{LTP}}$), the network is learning the SR with TD($\lambda \sim 0$). For quick sequential activities (T $\rightarrow$ 0), akin to hippocampal replays, the network is learning the SR with TD($\lambda \sim 1$). As we will discuss below, this framework therefore combines learning based on rate coding as well as temporal coding. Furthermore, from our model follows the prediction that replays can also be used for learning purposes and that they are algorithmically equivalent to MC, whereas during behavior, the hippocampal learning algorithm is equivalent to TD($\lambda$). This strategy of using replays to learn is in line with recent experimental and theoretical observations (see *Momennejad, 2020* for a review).

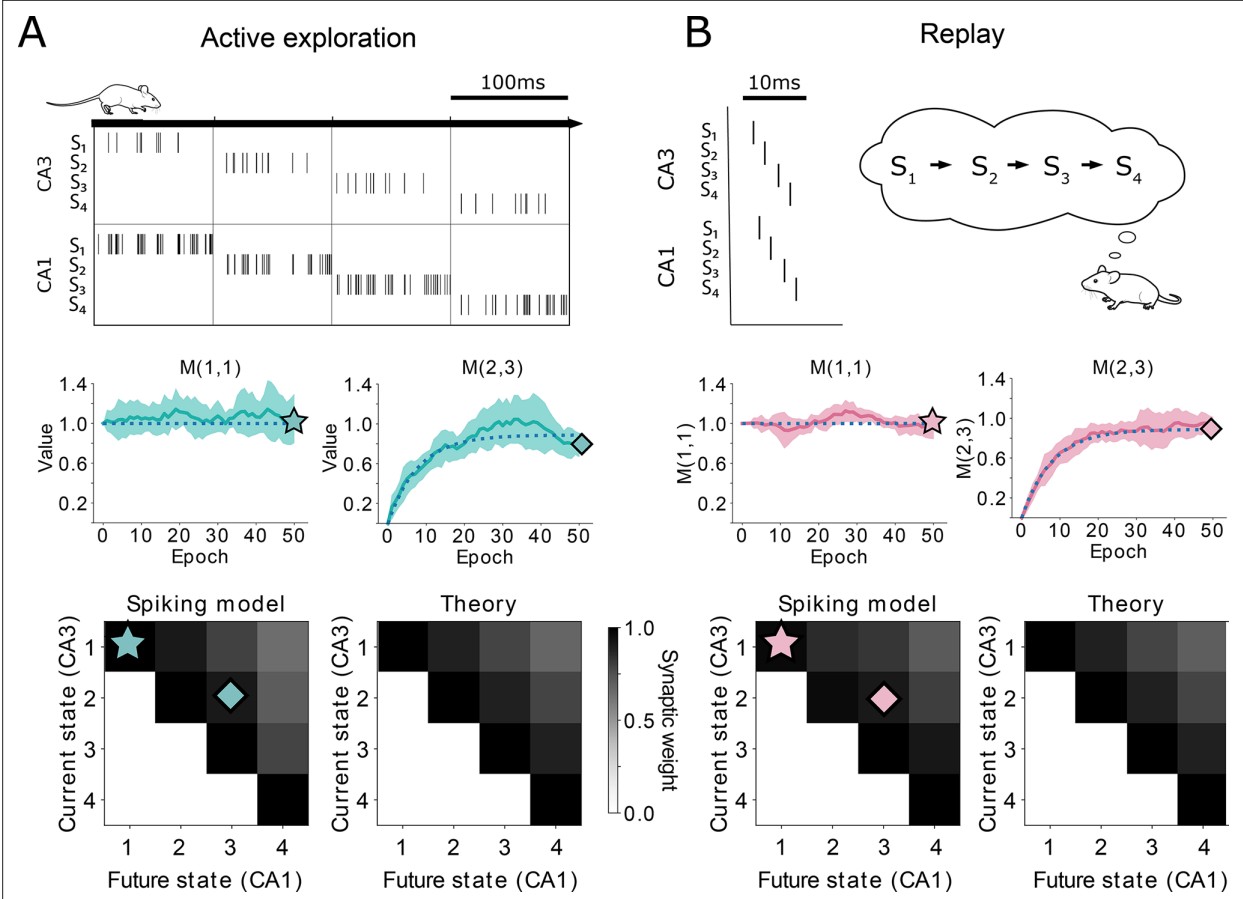

**Figure 2.** Comparison between TD($\lambda$) and our spiking model. (**A-top**) Learning during behavior corresponds to TD($\lambda \approx 0$). States are traversed on timescales larger than the plasticity timescales and place cells use a rate-code. (**A-middle**) Comparison of the learning over epochs for two synaptic connections (full line denotes the mean over ten random seeds, shaded area denotes one standard deviation) with the theoretical learning curve of TD($\lambda$) (dotted line). (**A-bottom**) Final successor matrix learned by the spiking model (left) and the theoretical TD($\lambda$) algorithm (right). Star and diamond symbols denote the corresponding weights shown in the middle row. (B-top) Learning during replays corresponds to TD($\lambda \approx 1$). States are traversed on timescales similar to the plasticity timescales and place cells use a temporally precise code. (**B-middle and bottom**) Analogous to panel A middle and bottom.

The online version of this article includes the following figure supplement(s) for figure 2:

**Figure supplement 1.** Comparison of the exact and approximate equations for the parameters.

To validate our analytical results, we use again a linear track with a deterministic policy. Using our spiking model with either rate-code activity on behavioral time-scales (*Figure 2a* top) or temporal-code activity similar to replays (*Figure 2b* top), we show that the synaptic weights across trials match the evolution of the TD($\lambda$) algorithm closely (*Figure 2a and b* middle). While convergence to the SR is guaranteed (*Figure 2a and b* bottom) due to the mathematical equivalence between our setup and TD($\lambda$) (*Figure 2—figure supplement 1*), the learning trajectory has more variance in the neural network case due to the noise introduced by the randomness of the spike times. This noise can be mitigated by averaging over a population of neurons. Moreover, due to the equivalence with TD($\lambda$), our setup is general for any type of task where discrete states are visited, in any dimension, and which may not need to be a navigation task (see e.g. *Figure 1—figure supplement 1* for a 2D environment).

In summary, we showed how the network can learn the SR using a spiking neural model. We analytically showed how the learning algorithm is equivalent to TD($\lambda$), and confirmed this using numerical simulations. We derived a relationship between the abstract parameter $\lambda$ and the timescale T representing the animal's behavior — and in turn the neuronal spiking — allowing us to unify rate and temporal coding within one framework. Furthermore, we predict a role for hippocampal replays in learning the SR using an algorithm equivalent to Monte Carlo.

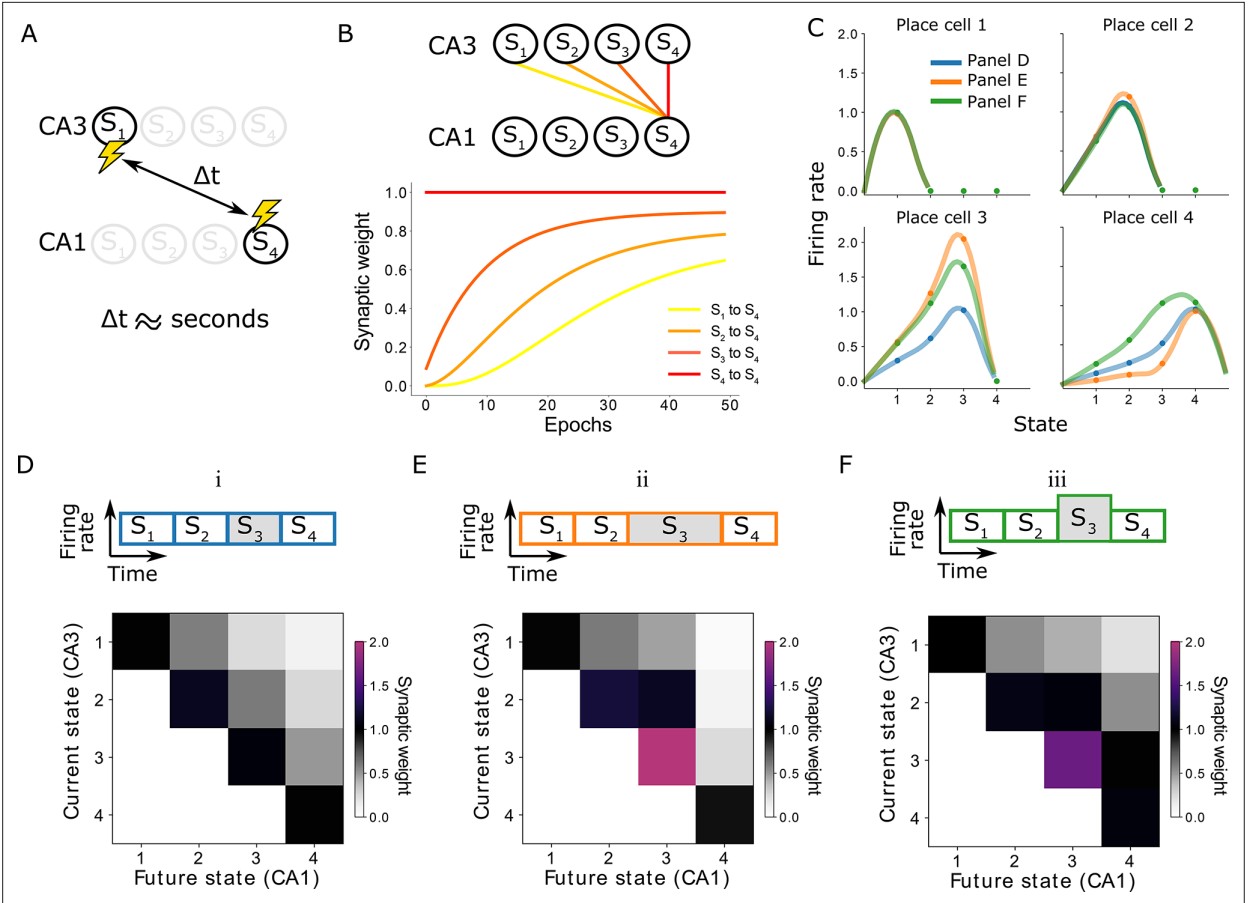

**Figure 3.** Learning on behavioral timescales and state-dependent discounting. (**A**) In our model, the network can learn relationships between neurons that are active seconds apart, while the plasticity rule acts on a millisecond timescale. (**B**) Due to transitions between subsequent states, each synaptic weight update depends on the weight from the subsequent CA3 neuron to the same CA1 neuron. In other words, the change of a synaptic weight depends on the weight below it in the successor matrix. The top panel visualizes how weights depend on others in our linear track example, where each lighter color depends on the darker neighbor. The bottom panel shows the learning of over 50 epochs. Notice the lighter traces converge more slowly, due to their dependence on the darker traces. (**C**) Place fields of the place cells in the linear track — each place cell corresponding to a column of the successor matrix. Activities of each place cell when the animal is in each of the four states (dots) are interpolated (lines). Three variations are considered: (i) the time spent in each state and the CA3 firing rates are constant (blue and panel **D**); (ii) the time spent in state 3 is doubled (orange and panel **E**); (iii) the CA3 firing rate in state 3 is doubled (green and panel **F**). Panels E and F lead to a modified discount parameter in state 3, affecting the receptive fields of place cells 3 and 4.

## Learning over behavioral time-scales using STDP

An important observation in our framework is that the SR can be learned *using the same underlying STDP rule* over time-scales ranging from replays up to behavior. One can now wonder how it is possible to learn relationships between events that are seconds apart during awake behavior, without any explicit error encoding signal typically used by the TD algorithm, and while the STDP rule is characterised by millisecond time-scales (*Figure 3a*).

From a neuroscience perspective, this can be understood when considering the trajectory of the animal. Each time the animal moves from a position $S_{j-1}$ to a position $S_j$, the CA3 cell encoding the location $S_{j-1}$ stops firing and the CA3 cell encoding the location $S_j$ starts firing. Since in our example this transition is instantaneous, these cells are activating the same CA1 cells consecutively. Therefore, the change in the weight $w_{i,j-1}$ depend on the synaptic weight of the subsequent state $w_{i,j}$ (*Figure 3b*, yellow depends on orange, orange depends on red, etc). Indeed, in our example of an animal in a linear track subdivided into four locations, the weights on the diagonal, such as $w_{4,4}$, are the first ones to be learned, since they are learned directly. The off-diagonal weights, such as $w_{3,4}$, $w_{2,4}$, and $w_{1,4}$, are learned consecutively more slowly as they are dependent on the subsequent synaptic weight.

Eventually, weights between neurons encoding positions that are behaviorally far apart can be learnt using a learning rule on a synaptic timescale (*Figure 3b*).

From a reinforcement learning perspective, the TD(0) algorithm relies on a property called bootstrapping. This means that the successor representation is learned by first taking an initial estimate of the SR matrix (i.e. the previously learned weights), and then gradually adjusting this estimate (i.e. the synaptic weights) by comparing it to the states in the environment that the animal actually visits. This comparison is achieved by calculating a *prediction error*, similar to the widely studied one for dopamine neurons (*Schultz et al., 1997*). Since the synaptic connections carry information about the expected trajectories, in this case, the prediction error is computed between the predicted and observed trajectories (see Materials and methods).

The main point of bootstrapping, therefore, is that learning happens by adjusting our current predictions (e.g. synaptic weights) to match the observed current state. This information is available at each time step and thus allows learning over long timescales using synaptic plasticity alone. If the animal moves to a state in the environment that the current weights deem unlikely, potentiation will prevail and the weight from the previous to the current state will increase. Otherwise, the opposite will happen. It is important to notice that the prediction error in our model is not encoded by a separate mechanism in the way that dopamine is thought to do for reward prediction (*Schultz et al., 1997*). Instead, the prediction error is represented locally, at the level of the synapse, through the depression and potentiation terms of our STDP rule, and the current weight encodes the current estimate of the SR (see Materials and methods). Notably, the prediction error is equivalent to the TD($\lambda$) update. This mathematical equivalence ensures that the weights of our neural network track the TD($\lambda$) update at each state, and thus stability and convergence to the theoretical values of the SR. We therefore do not need an external vector to carry prediction error signals as proposed in *Gardner et al., 2018*; *Gershman, 2018*. In fact, the synaptic potentiation in our model updates a row of the SR, while the synaptic depression updates a column.

On the other extreme, for very fast timescales such as replays, TD(1) is equivalent to online Monte Carlo learning (MC), which does not bootstrap at all. Instead, MC samples the whole trajectory and then simply takes the average of the discounted state occupancies to update the SR (see Materials and methods). During replays, the whole trajectory falls under the plasticity window and the network can learn without bootstrapping. For all cases in between, the network partially relies on bootstrapping and we correspondingly find a $\lambda$ between 0 and 1.

In summary, in our framework, synaptic plasticity leads to the development of a successor representation in which synaptic weights can be directly linked to the successor matrix. In this framework, we can learn over behavioral timescales even though our plasticity rule acts on the scale of milliseconds, due to the bootstrapping property of TD algorithms.

## Different discounting for space and time

In reinforcement learning, it is usual to have delay-discounting: rewards that are further away in the future are discounted compared to rewards that are in the immediate future. Intuitively, it is indeed clear that a state leading to a quick reward can be regarded as more valuable compared to a state that only leads to an equal reward in the distant future. For tasks in a tabular setting, with a discrete state space and where actions are taken in discrete turns, such as for example chess or our simple linear track discussed in section 'The Successor Representation', one can simply use a multiplicative factor $0 < \gamma \leq 1$ for each state transition. In this case the discount follows an exponential dependence, where rewards that are $n$ steps away are discounted by a factor of $\gamma^n$.

In order to still use the above exponential discount when time is continuous, the usual approach is to discretize time by choosing a unit of time. However, this would imply one can never remain in a state for a fraction of this unit of time, and it is not clear how this unit would be chosen. Our framework deals naturally with continuous time, through the monotonically decreasing dependence of the discount parameter $\gamma$ on the time an agent remains in a state, T. The dependence on T can be interpreted as an increased discounting the longer a state lasts.

In this way, instead of discounting by $\gamma^n$ when the agent stays $n$ units of time in a certain state, we would discount by $\gamma(n \cdot T)$. More generally, for any arbitrary time T, a discount corresponding to $\gamma(T)$ will be applied. This allows the agent to act in continuous time (*Figure 3c and e*). Interestingly, the dependence of $\gamma$ on T in our model is not exponential as in the tabular case. Instead, we have a

hyperbolic dependence. This hyperbolic discount is well studied in psychology and neuroeconomics and appears to agree well with experimental results (*Laibson, 1997*; *Ainslie, 2012*).

The difference between a hyperbolic discount and an exponential discount lays in the fact that we will attribute a different value to the same temporal delay, depending on whether it happens sooner or later. A classic example is that, when given the choice, people tend to prefer 100 dollars today instead of 101 dollars tomorrow, while they tend to prefer 101 dollars in 31 days instead of 100 dollars in 30 days. They therefore judge the 1 day of delay differently when it happens later in time. Exponential discounting, on the other hand, always attributes the same value to the same delay no matter when it occurs.

Our model therefore combines two types of discounting: exponential when we move through space — when sequentially activating different place cells — and hyperbolic when we move through time — when we prolong the activity of the same place cell.

The discount factor $\gamma$ also depends on other parameters such as firing rate and STDP amplitudes (see *Equation 22* in the Appendix). This gives our model the flexibility to encode state-dependent discounting even when the trajectories and times spent in the states are the same. Such state-dependent discounting can be useful to for example encode salient locations in the environment such as landmarks or reward locations (*Figure 3c and f*).

## Bias-variance trade-off

As discussed previously (section 'Learning the successor representation in biologically plausible networks'), the TD($\lambda$) algorithm unifies the TD algorithm and the MC algorithm. In our framework, replay-like neuronal activations are equivalent to MC, while behavioral-like activity is equivalent to TD. In this section, we will discuss how the replays and behavior can work together when learning the cognitive map of an environment, leveraging the strengths of MC and TD.

The MC algorithm effectively works by averaging over the sampled trajectories. As such, the estimated SR matrix will be a close approximation of the theoretical value. The difference between the estimated and theoretical value is commonly referred to as bias. We can therefore say that the MC algorithm presents low bias. However, if the agent moves in the environment at random, the sampled trajectories will be quite different from each other. When taking the average, the estimated value will therefore fluctuate a lot. In this case, we say that the MC estimate has high variance as well (*Figure 4A and B*).

Unlike MC, the TD algorithm updates its estimate of the SR by comparing the current estimate of the SR with the actual state the agent transitioned to. Because of the dependence on the current estimate, this estimate will be incrementally refined with small updates. In this way, the SR estimate will not fluctuate much, and be lower in variance. However, by this dependence on the current estimate, we introduce a bias in the algorithm, which will be especially significant when our initial estimate of the SR is bad (*Figure 4A and B*). The TD algorithm therefore presents high bias and low variance.

We now apply these concepts to learning in a novel environment. Since the MC algorithm is unbiased by the initial estimate of the SR, replays should initially speed up learning in an unfamiliar environment. Later on, when the environment becomes familiar, the SR estimate is already closer to the exact value. At this point, we prefer to have low variance and thus the TD algorithm will be preferred. We confirm this logic using our spiking neural networks, and show how we can have both quick learning and low error at convergence if we proportionally have more replays at the first trials in a novel environment (*Figure 4a–e*). In contrast, when having an equal proportion of replays throughout the whole simulation, we do not yield as quick learning as MC and as low asymptotic error as TD (*Figure 4—figure supplement 1*). Interestingly, the pattern of proportionally more replays in novel environments versus familiar environments has also been experimentally observed (*Cheng and Frank, 2008*; *Figure 4f*). Please note that, while we implemented an exponentially decaying probability for replays after entering a novel environment, different schemes for replay activity could be investigated. Note also that other mechanisms besides the successor representation could account for these results, including model-based reinforcement learning.

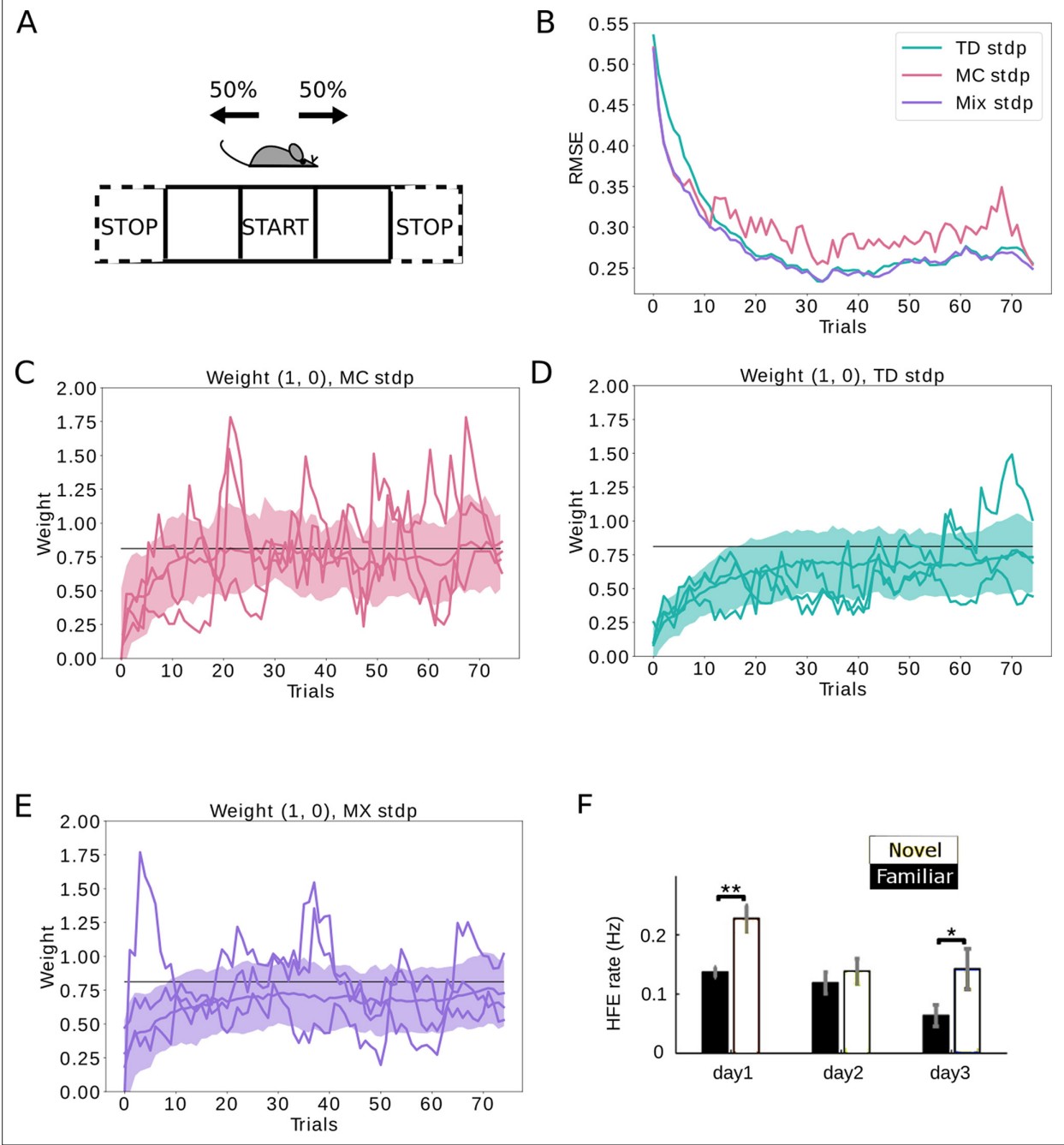

**Figure 4.** Replays can be used to control the bias-variance trade-off. (**A**) The agent follows a stochastic policy starting from the initial state (denoted by *START*). The probability to move to either neighboring state is 50%. An epoch stops when reaching a terminal state (denoted with *STOP*). (**B**) Root mean squared error (RMSE) between the learned SR estimate and the theoretical SR matrix. The full lines are mean RMSEs over 1000 random seeds. Three cases are considered: (i) learning happens exclusively due to behavioral activity (TD STDP, green); (ii) learning happens exclusively due to replay activity (MC STDP, purple); (iii) A mixture of behavioral and replay learning, where the probabilities for replays drops off exponentially with epochs (Mix STDP, pink). The *mix* model, with a decaying number of replays learns as quickly as MC in the first epochs and converges to a low error similar to TD, benefiting both from the low bias of MC at the start and the low variance of TD at the end. (**C, D, E**) Representative weight changes for each of the scenarios. Full lines show various random seeds, shaded areas denote one standard deviation over 1000 random seeds. (**F**) More replays are observed when an animal explores a novel environment (day 1). Panel F adapted from Figure 3A in ***Cheng and Frank, 2008***.

The online version of this article includes the following figure supplement(s) for figure 4:

**Figure supplement 1.** Combining equal amounts of replays and behavioral learning.

**Figure supplement 2.** Setting the noise for replays.

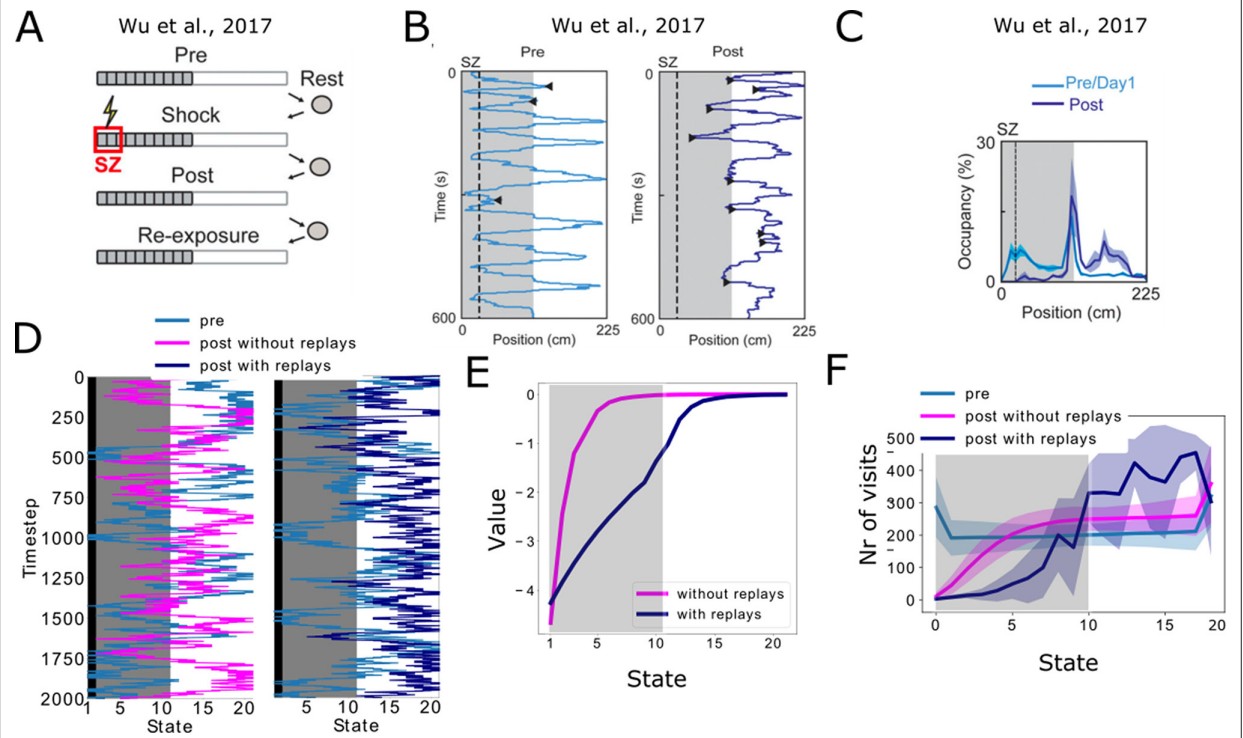

**Figure 5.** Reproducing place-avoidance experiments with the spiking model. (**A–C**) Data from *Wu et al., 2017*. (**A**) Experimental protocol: the animal is first allowed to freely run in the track (Pre). In the next trial, a footshock is given in the shock zone (SZ). In subsequent trials the animal is again free to run in the track (Post, Re-exposure). Figure redrawn from *Wu et al., 2017*. (**B**) In the Post trial, the animal learned to avoid the shock zone completely and the also mostly avoids the dark area of the track. Figure redrawn from *Wu et al., 2017*. (**C**) Time spent per location confirms that the animal prefers the light part of the track in the Post trial. Figure redrawn from *Wu et al., 2017*. (**D**) Mimicking the results from *Wu et al., 2017*, the shock zone is indicated by the black region, the dark zone by the gray region and the light zone by the white region. Left: without replays, the agent keeps extensively exploring the dark zone even after having experienced the shock. Right: with replays, the agent largely avoids entering the dark zone after having experienced the shock (replays not shown). (**E**) The value of each state in the cases with and without replays. (**F**) Occupancy of each state in our simulations and for the various trials. Solid line and shaded areas denote the average and standard deviation over 100 simulations, respectively. Notice we do not reproduce the peak of occupancy at the middle of the track as seen in panel c, since our simplified model assumes the same amount of time is spent in each state.

The online version of this article includes the following figure supplement(s) for figure 5:

**Figure supplement 1.** Doubling the time-steps in the scenario without replays.

**Figure supplement 2.** The value of states can be read out by downstream neurons.

**Figure supplement 3.** Dependency of $\gamma$, $\lambda$ and the place-tuned input to CA1 on $\theta/T$, for various values of T and depression amplitude $A_{pre}$.

**Figure supplement 4.** Readout of the state value for various parameters.

## Leveraging replays to learn novel trajectories

In the previous section, the replays re-activated the same trajectories as seen during behavior. In this section, we extend this idea and show how in our model replays can be useful during learning even when the re-activated trajectories were not directly experienced during behavior.

For this purpose, we reproduce an place-avoidance experiment from *Wu et al., 2017*. In short, rats are allowed to freely explore a linear track on day 1. Half of the track is dark, while the other half is bright. On day 2, the animals did four trials separated by resting periods: in the first trial (pre), the animals were free to explore the track; in the second trial (shock), they started in the light zone and received two mild footshocks when entering in the shock zone; in the third and fourth trial (post and re-exposure, respectively), they were allowed to freely explore the track again, but starting from the light zone or the shock zone respectively (*Figure 5a*). In the study, it was reported that during the post trial, animals tended to stay in the light zone and forward replays from the current position to the shock zone were observed when the animals reached the boundary between the light and the dark zone (*Figure 5b and c*).

We simulated a simplified version of this task. Our simulated agent moves through the linear track following a softmax policy, and all states have equal value during the first phase (pre) (*Figure 5d*, blue trajectories). Then, the agent is allowed to move through the linear track until it reaches the shock zone and experiences a negative reward. Finally, the third phase is similar as the first phase and the animal is free to explore the track. Two versions of this third phase were simulated. In one version, there are no replays (*Figure 5d*, orange trajectories in left panel), while in the second version a forward replay until the shock zone is simulated every time the agent enters the middle state (*Figure 5d*, orange trajectories in right panel, replays not shown). The replays affect the learning of the successor representation and the negative reward information is propagated towards the decision point in the middle of the track. The states in the dark zone therefore have lower value compared to the case without replays (*Figure 5e*). In turn, this different value affects the policy of the agent which now tends to avoid the dark zone all together, while the agent without replays still occupies many states of the dark zone as much as states in the light zone (*Figure 5f*). Moreover, even when doubling the amount of SR updates in the scenario without replays, the behavior of the agent remains unaltered (*Figure 5—figure supplement 1*). This shows that it is not the amount of updates, but the type of policy that is important when updating the SR, and how using a different policy in the replay activity can significantly alter behavior.

Our setup for this simulation is simplified, and does not aim to reproduce the complex decision making of the rats. Observe for example the peak of occupancy of the middle state by the animals (*Figure 5c*), which is not captured by our model because we assume the agent to spend the same amount of time in each state. Nonetheless, it is interesting to see how replaying trajectories that were not directly experienced before, in combination with a model allowing replays to affect the learning of a cognitive map, can substantially influence the final policy of an agent and the overall performance. This mental imagination of trajectories could be exploited to refine our cognitive maps, avoiding unfavourable locations or finding shortcuts to rewards. It is important to note here that, while we are suggesting a potential role for the SR in solving this task, the data itself would also be compatible with a model-based strategy. In fact, experimental evidence suggests that humans may use a mixed strategy involving both model-based reinforcement learning and the successor representation (*Momennejad et al., 2017*).

## Discussion

In this article, we investigated how a spiking neural network model of the hippocampus can learn the successor representation. Interestingly, we show that the updates in synaptic weights resulting from our biologically plausible STDP rule are equivalent to TD($\lambda$) updates, a well-known and powerful reinforcement learning algorithm.

### Reinforcement learning

Our network learns the SR in the CA3-CA1 weights. Since we have modeled neurons to integrate the synaptic EPSPs and generate spikes using an inhomogeneous Poisson process based on the depolarization, the firing rate is proportional to the total synaptic weights. Therefore, the successor representation can be read out simply by a downstream neuron. Moreover, since the value of a state is defined by the inner product between the successor matrix and the reward vector, it is sufficient for the synaptic weights to the downstream neuron to learn the reward vector, and the downstream neuron will then encode the state value in its firing rate (see *Figure 5—figure supplement 2*). While the neuron model used is simple, it will be interesting for future work to study analogous models with non-linear neurons.

It is worth noting that, during learning, both pre-synaptic and post-synaptic layers receive external inputs representing the current state (*Equation 10* and *Equation 11* in Materials and methods). This may induce a distortion in the read out of the diagonal elements of the SR matrix (see *Equations 13 and 15*, and *Figure 5—figure supplement 2*). At a first glance, this may indicate that learning and reading out are antagonistic. However, there are multiple ways we could resolve this apparent conflict: (i) Since the external current in CA1 is present for only a fraction of the time T in each state, the readout might happen during the period of CA3 activation exclusively; (ii) The readout may be over the whole time T but becomes more noisy towards the end. Even in the case where the readout is

noisy, the distortion would be limited to the diagonal elements of the matrix; (iii) Learning and readout may be separate mechanisms, where the CA3 driving current is present during readout only. This could be for instance signaled by neuromodulation (e.g. noradrenaline and acetylcholine are active during learning but not exploration *Micheau and Marighetto, 2011*; *Hasselmo and Sarter, 2011*; *Robbins, 1997*; *Teles-Grilo Ruivo and Mellor, 2013*; *Palacios-Filardo et al., 2021*), or it could be that readout happens during replays; (iv) The weights to or activation functions of the readout neuron may learn to compensate for the distorted signal in CA1.

Furthermore, we can notice that the external inputs encoding the current state activate CA3 first, and CA1 later. The delay between these activations $\theta/T$ (*Equation 10* and *Equation 11* in Materials and methods) is an arbitrary parameter that can be adjusted. Varying this delay will change the reinforcement learning representation, especially parameters $\lambda$ and $\gamma$, but also the strength of the input current (see *Figure 5—figure supplement 3*). However, this will not impact the distortion of the diagonal elements of the SR matrix, which remains similar across various delay values $\theta/T$ (see *Figure 5—figure supplement 4*).

## Biological plausibility

Uncovering a connection between STDP and TD($\lambda$) shows how, using minimal assumptions, a theoretically grounded learning algorithm can emerge from a biological implementation of plasticity. Similar learning rules have indeed been observed in the hippocampus (*Shouval et al., 2002* and proposed on theoretical grounds *Mehta et al., 2000*; *Waddington et al., 2012*; *van Rossum et al., 2012*).

The TD algorithm is most commonly known in neuroscience for describing how reward prediction can be computed in the brain. More specifically, it is widely believed that dopamine neurons in the ventral tegmental area (VTA) and substantia nigra (SNc) encode the prediction error between the observed and expected reward (*Schultz et al., 1997*), dopamine thus acts as a global signal that can be broadcasted to other areas of the brain like the striatum to compute the expected reward. In our model, the TD algorithm estimates the SR (i.e. expected future occupancy), rather than the value. However, since the prediction error for the SR is different for every synaptic connection (i.e. each pair of states), it is not clear how it could be carried by a global signal analogous to dopamine. The SR would need multiple signals, or a matrix transformation of the global signal. Furthermore, we would need to postulate that such error – or errors – are computed elsewhere in the brain. Instead, in our model, the prediction error simply emerges from the synaptic plasticity rule itself. Furthermore, thanks to the presynaptic depression, our STDP rule alone allows us to compute negative prediction errors, which still poses an open challenge for computation with dopamine because of the low baseline dopaminergic firing rate (*Glimcher, 2011*; *Daw et al., 2002*; *Matsumoto and Hikosaka, 2007*).

Our framework smoothly connects a temporally precise spiking code with a fully rate-based code, and anything in between. As we have proven mathematically, this translates in moving smoothly from Monte Carlo to Temporal Difference by means of TD($\lambda$). Fast spiking sequences (temporal code) can be used for consolidation of previous experiences using Monte Carlo learning, while the behavioral timescale activity (rate code) results in TD updates, allowing learning on the timescale of seconds even with plasticity timeconstants on the order of milliseconds. This type of Hebbian learning over behavioral timescale exploits the bootstrapping property of TD, and is different than the one-shot behavioral plasticity described in *Bittner et al., 2017*. However, these two mechanisms could be complementary, where the latter could play a more significant role in the formation of new place fields, while the former would be more relevant to shape the existing place fields to contain predictive information. Learning on behavioral timescales using STDP was also investigated in *Drew and Abbott, 2006*. The main difference between *Drew and Abbott, 2006* and our work, is that the former relies on overlapping neural activity between the pre- and post-synaptic neurons from the start, while in our case no such overlap is required. In other words, our setup allows us to learn connections between a presynaptic neuron and a postsynaptic neuron whose activities are separated by behavioral timescales initially. For this to be possible, there are two requirements: (1) the task needs to be repeated many times and (2) a chain of neurons are consecutively activated between the aforementioned presynaptic and postsynaptic neuron. Due to this chain of neurons, over time the activity of the postsynaptic neuron will start earlier, eventually overlapping with the presynaptic neuron.

In our work, we did not include theta modulation, but phase precession and theta sequences could be yet another type of activity within the TD lambda framework. A recent work (*George et al.,*

*2023*) incorporated the theta sweeps into behavioral activity, showing it approximately learns the SR. Moreover, theta sequences allow for fast learning, playing a similar role as replays (or any other fast temporal-code sequences) in our work. By simulating the temporally compressed and precise theta sequences, their model also reconciles the learning over behavioral timescales with STDP. In contrast, our framework reconciles both timescales relying purely on rate-coding during behavior. Finally, their method allows to learn the SR within continuous space. It would be interesting to investigate whether these methods co-exist in the hippocampus and other brain areas. Furthermore, (*Fang et al., 2023*) et al. recently showed how the SR can be learned using recurrent neural networks with biologically plausible plasticity.

There are three different neural activities in our proposed framework: the presynaptic layer (CA3), the postsynaptic layer (CA1), and the external inputs. These external inputs could for example be location-dependent currents from the entorhinal cortex, with timings guided by the theta oscillations. The dependence of CA1 place fields on CA3 and entorhinal input is in line with lesion studies (see e.g. *Brun et al., 2008*; *Hales et al., 2014*; *O'Reilly et al., 2014*). It would be interesting for future studies to further dissect the role various areas play in learning cognitive maps.

Notably, even though we have focused on the hippocampus in our work, the SR does not require predictive information to come from higher-level feedback inputs. This framework could therefore be useful even in sensory areas: certain stimuli are usually followed by other stimuli, essentially creating a sequence of states whose temporal structure can be encoded in the network using our framework. Interestingly, replays have been observed in other brain areas besides the hippocampus (*Kurth-Nelson et al., 2016*; *Staresina et al., 2013*). Furthermore, temporal difference learning in itself has been proposed in the past as a way to implement prospective coding (*Brea et al., 2016*).

## Replays

We have also proposed a role for replays in learning the SR, in line with experimental findings and RL theories (*Russek et al., 2017*; *Momennejad et al., 2017*). In general, replays are thought to serve different functions, spanning from consolidation to planning (*Roscow et al., 2021*). Here, we have shown that when the replayed trajectories are similar to the ones observed during behavior, they play the role of speeding up and consolidating learning by regulating the bias-variance trade-off, which is especially useful in novel environments. On the other hand, if the replayed trajectories differ from the ones experienced during wakefulness, replays can play a role in reshaping the representation of space, which would suggest their involvement in planning. Experimentally, it has been observed that replays often start and end from relevant locations in the environment, like reward sites, decision points, obstacles or the current position of the animal (*Ólafsdóttir et al., 2015*; *Pfeiffer and Foster, 2013*; *Jackson et al., 2006*; *Mattar and Daw, 2017*). Since these are salient locations, it is in line with our proposition that replays can be used to maintain a convenient representation of the environment. It is worth noticing that replays can serve a variety of functions, and our framework merely proposes additional beneficial properties without claiming to explain all observed replays. For example, in addition to forward replays, also reverse replays are ubiquitous in hippocampus (*Pfeiffer, 2020*). The reverse replays are not included in our framework, and it is not clear yet whether they play different roles, with some evidence suggesting that reverse replays are more closely tied to the reward encoding (*Ambrose et al., 2016*). Moreover, while indirect evidence supports the idea that replays can play a role during learning (*Igata et al., 2021*), it is not yet clear how synaptic plasticity is manifested during replays (*Fuchsberger and Paulsen, 2022*).

## Learning flexibility

Multiple ideas from reinforcement learning, such as TD($\lambda$), state-dependent discounting and the successor representation, emerge quite naturally from our simple biologically plausible setting. We propose in our work that time and space can be discounted differently. Moreover, the flexibility to change the discounting factor by modulating firing rates and plasticity parameters — which is ubiquitous in neural circuits — suggests that these mechanisms could be used to encode a variety of information in a cognitive map. Moreover, the specific dependence of the discount factor on the biological parameters leads to experimentally testable predictions. Indeed, our framework predicts well-defined changes in place fields after modulations of firing rates, speed of the agent or neuromodulation of the plasticity parameters (*Figure 3*). Importantly, the discount parameter also depends on the time spent

in each state. This eliminates the need for time discretization, which does not reflect the continuous nature of the response of time cells (*Kraus et al., 2013*).

## Limitations of the reinforcement learning framework

We have already outlined some of the benefits of using reinforcement learning for modeling behavior, including providing clear computational and algorithmic frameworks. However, there are several intrinsic limitations to this framework. For example, RL agents that only use spatial data do not provide complete descriptions of behavior, which likely arises from integrating information across multiple sensory inputs. Whereas an animal would be able to smell and see a reward from a certain distance, an agent exploring the environment would only be able to discover it when randomly visiting the exact reward location. Furthermore, the framework rests on fairly strict mathematical assumptions: typically the state space needs to be markovian, time and space need to be discretized (which we manage to evade in this particular framework) and the discounting needs to follow an exponential decay. These assumptions are simplistic and it is not clear how often they are actually met. Reinforcement Learning is also a sample-intensive technique, whereas we know that some animals, including humans, are capable of much faster or even one-shot learning.

Even though we have provided a neural implementation of the SR, and of the value function as its read-out (see *Figure 5—figure supplement 2*), the whole action selection process is still computed only at the algorithmic level. It may be interesting to extend the neural implementation to the policy selection mechanism in the future.

Taken together, our work joins — in a single framework — a variety of concepts from the neuronal level over cognitive theories to reinforcement learning.

## Materials and methods

### The successor representation

In a tabular environment, we define the value of a state $s$ as being the expected cumulative reward that an agent will receive following a certain policy starting in $s$. The future rewards are multiplied by a factor $0 < \gamma^n \leq 1$, where $n$ is the number of steps until reaching the reward location and $0 < \gamma \leq 1$ is the delay discount factor. It is usual to use $0 < \gamma < 1$, which ensures that earlier rewards are given more importance compared to later rewards. Formally, the value of a state $s$ under a certain policy $\pi$ is defined as

$$V^{\pi}(s) \quad = \mathbb{E}_{\pi}\left[\sum_{k=0}^{\infty} \gamma^k R_{t+k} \,\middle|\, S_t = s\right] \tag{1}$$

$$= \sum_a \pi(a|s)\left[R(s, a) + \sum_{s'} P(s'|s, a)\gamma V^{\pi}(s')\right] \tag{2}$$

Here, $a$ denotes the action, $R(s, a)$ is the reward function and $P(s'|s, a)$ is the transition function, i.e. the probability that taking an action $a$ in state $s$ will result in a transition to state $s'$. Following (*Dayan, 1993*), we can decompose the value function into the inner product of reward function and successor matrix

$$V(s) = \sum_{s'} M_{s,s'} R(s') \tag{3}$$

with

$$M_{s,s'} = \mathbb{E}\left[\sum_{t=0}^{\infty} \gamma^t \mathbb{I}(s_t = s') \,\middle|\, s_0 = s\right] \tag{4}$$

This representation is known as the successor representation (SR), where each element $M_{ij}$ represents the expected future occupancy of state $j$ when in state $i$. By decomposing the value into the SR and the reward function (*Equation 3*), relearning the state values $V$ after changing the reward function is fast, similar to model-based learning. At the same time, the SR can be learned in a model-free manner, using for example temporal difference (TD) learning (*Russek et al., 2017*).

### Derivation of the TD($\lambda$) update for the SR

The TD($\lambda$) update for the SR is then implemented according to (see e.g. *Sutton and Barto, 1998*)

$$\Delta M(j, i) \quad = \delta_0^{TD} + \gamma\lambda\delta_1^{TD} + (\gamma\lambda)^2\delta_2^{TD} + \dots \tag{5}$$

Using $\delta_i^{TD}$ for the TD error at step $i$ and $\delta_{xy}$ for the Kronecker delta,

$$\delta_n^{TD} = \delta_{j+n,i+n} + \gamma M(j+n+1, i+n) - M(j+n, i+n) \tag{6}$$

corresponds to the TD error for element $M(j+n, i+n)$ of the successor representation after the transition from state $j+n$ to state $j+n+1$. Combining *Equations 5 and 6*, we find

$$
\begin{aligned}
\Delta M(j, i) \quad &= [\delta_{j,i} + \gamma M(j+1, i) - M(j, i)] \\
&\quad + \gamma\lambda[\delta_{j+1,i} + \gamma M(j+2, i) - M(j+1, i)] \\
&\quad + (\gamma\lambda)^2[\delta_{j+2,i} + \gamma M(j+3, i) - M(j+2, i)] \\
&\quad + \dots \\
&= -M(j, i) + \delta_{j,i} + (1-\lambda)\gamma M(j+1, i) + \gamma\lambda\delta_{j+1,i} \\
&\quad + (1-\lambda)\lambda\gamma^2 M(j+2, i) + (\gamma\lambda)^2\delta_{j+2,i} + \dots \\
&= -M(j, i) + \sum_{n=0}^{N}[(\gamma\lambda)^n\delta_{j+n,i} + (1-\lambda)\gamma(\gamma\lambda)^n M(j+n+1, i)]
\end{aligned}
\tag{7}
$$

and

$$
\begin{aligned}
M(j, i) \quad &\leftarrow M(j, i) + \eta\,\Delta M(j, i) \\
&\leftarrow M(j, i) - \eta\{M(j, i) + \sum_{n=0}^{N}[(\gamma\lambda)^n\delta_{j+n,i} + (1-\lambda)\gamma(\gamma\lambda)^n M(j+n+1, i)]\}
\end{aligned}
\tag{8}
$$

## Neural network model

### Plasticity rule

The synaptic plasticity rule (*Figure 1d*) consists of a weight-dependent depression for presynaptic spikes and a spike-timing dependent potentiation, given by

$$
\begin{aligned}
\frac{dw_{ij}(t)}{dt} &= \eta_{\text{STDP}}A_{\text{LTP}} \cdot Tr_{LTP}^j(t) \cdot \sum_i \delta(t - t^i) - \eta_{\text{STDP}}A_{\text{LTD}} \cdot w_{ij}(t) \cdot \delta(t - t^j) \\
\tau_{\text{LTP}}\frac{dTr_{LTP}^j(t)}{dt} &= -Tr_{LTP}^j(t) + \sum_j \delta(t - t^j)
\end{aligned}
\tag{9}
$$

Here, $w_{ij}$ represents the synaptic connection from presynaptic neuron $j$ to postsynaptic neuron $i$, $Tr_{LTP}^j$ is the plasticity trace, a low-pass filter of the presynaptic spike train with time constant $\tau_{\text{LTP}}$, $t^j$ and $t^i$ are the spike times of the postsynaptic and presynaptic neuron respectively, $A_{\text{LTP}}$ and $A_{\text{LTD}}$ are the amplitudes of potentiation and depression respectively, $\eta_{\text{STDP}}$ is the learning rate for STDP and the $\delta(\cdot)$ denotes the Dirac delta function.

### Place cell activation

We assume that each state in the environment is represented by a population of place cells in the network. In our model, this is achieved by delivering place-tuned currents to the neurons. Whenever a state $S = j$ is entered, the presynaptic neurons encoding state $j$ start firing at a constant rate $\rho^{pre}$ for a time $\theta$, following a Poisson process with parameter $\rho_h^{pre}(t)$. The other presynaptic neurons are assumed to be silent:

$$
\rho_h^{pre}(t) = \begin{cases} \rho^{pre}\delta_{hj}, & \text{if } t \in [0, \theta) \\ 0 & \text{otherwise} \end{cases}
\tag{10}
$$

where the Kronecker delta function is defined as $\delta_{hj} = 1$ if $h = j$ and zero otherwise. Here we use the index $j$ to denote any neuron belonging to the population of neurons encoding state $j$. After a short

delay, at time $t^*$, a similar current $\rho^{bias}$ is delivered to the postsynaptic neuron encoding state $j$, for a duration of time $\omega$.

$$\rho_i^{bias}(t) = \begin{cases} \rho^{bias}\delta_{ij}, & \text{if } t \in [t^*, t^* + \omega) \\ 0, & \text{otherwise} \end{cases} \tag{11}$$

Besides the place-tuned input current, CA1 neurons receive inputs from the presynaptic layer (CA3). The postsynaptic potential $\rho_i^{post}$ when the agent is in state j is thus given by

$$\rho_i^{post}(t) = \sum_k^{N_{pop}} \sum_{t_k^f < t} w_{ij_k}(t)\kappa(t - t^f) + \rho_i^{bias}(t), \tag{12}$$

with the first sum running over all $N_{pop}$ presynaptic neurons encoding state $j$, and the second sum over all presynaptic firing times $t_k^f$ of neuron $k$ happened before $t$. The excitatory postsynaptic current $\kappa$ is modeled as an exponential decay described as $\kappa(x) = \epsilon_0 e^{-x/\tau_m}$ for $x \geq 0$ and zero otherwise. Each CA1 neuron $i$ fires following an inhomogeneous Poisson process with rate $\rho_i^{post}(t)$.

Note that, in most simulations we will use a single neuron in the population $N_{pop} = 1$. In addition, we normally set $t^* = \theta$ and $\omega = T - \theta$. However, we will keep these as explicit parameters for theoretical purposes.

## Equivalence with TD($\lambda$)
### Total plasticity update
Since we have the mathematical equation for the plasticity rule, and CA3 and CA1 neurons follow an inhomogeneous Poisson process with time-dependent firing rate, we can calculate analytically the average total weight change for the synapse $w_{ij}$, given a certain trajectory (details in the Appendix). Please notice that our calculation is based on *Kempter et al., 1999*, which takes into account the fact that our plasticity rule is sensitive to spike timing and involves a spike-spike correlation term. We find that:

$$\Delta w_{ij} = A\, w_{ij} + \sum_{n=0}^{N}[B(e^{-T/\tau_{LTP}})^n \delta_{ij+n} + C(e^{-T/\tau_{LTP}})^{n+1}w_{i,j+n+1}] \tag{13}$$

where $N$ is the number of states until the end of the trajectory and

$$A = \eta_{STDP}A_{LTP}\,N_{pop}\epsilon_0(\rho^{pre})^2\,\tau_{LTP}\,\tau_m(1 - e^{-\theta/\tau_m})\left[\theta - \tau_{LTP}(1 - e^{-\theta/\tau_{LTP}})\right]$$
$$+ \eta_{STDP}A_{LTP}\theta\rho^{pre}N_{pop}\frac{\tau_m\tau_{LTP}}{\tau_m+\tau_{LTP}}\epsilon_0 - \eta_{STDP}A_{pre}\,\rho^{pre}\theta \tag{14}$$

$$B = \eta_{STDP}A_{LTP}\rho^{pre}\tau_{LTP}^2(e^{\frac{\theta}{\tau_{LTP}}} - 1)e^{-\frac{t^*}{\tau_{LTP}}}(1 - e^{-\frac{\omega}{\tau_{LTP}}})\rho^{bias} = B'\rho^{bias} \tag{15}$$

$$C = \eta_{STDP}A_{LTP}N_{pop}\epsilon_0\tau_m\tau_{LTP}^2(\rho^{pre})^2(1 - e^{-\frac{\theta}{\tau_m}})(e^{\frac{\theta}{\tau_{LTP}}} - 1)(1 - e^{-\frac{\theta}{\tau_{LTP}}}) \tag{16}$$

### Comparison with TD($\lambda$)
Comparing the total weight change due to STDP (*Equation 13*) to the TD($\lambda$) update (*Equation 8*), we can see that the two equations are very similar in form:

$$w_{ij} \leftarrow w_{ij} - A\{-w_{ij} + \sum_{n=0}^{N}[-\frac{B}{A}(e^{-T/\tau_{LTP}})^n\delta_{ij+n} - \frac{C}{A}e^{-T/\tau_{LTP}}(e^{-T/\tau_{LTP}})^nw_{i,j+n+1}]\}$$

$$M(j, i) \leftarrow M(j, i) + \eta\{-M(j, i) + \sum_{n=0}^{N}[(\gamma\lambda)^n\delta_{j+n,i} + (1 - \lambda)\gamma(\gamma\lambda)^nM(j + n + 1, i)]\}$$

We impose $w_{ij} = M(j, i)$, and find:

$$-A = \eta \tag{17}$$

$$-\frac{B}{A} = 1 \quad \rightarrow \quad \rho^{bias} = -\frac{A}{B'} \tag{18}$$

$$e^{-T/\tau_{LTP}} = \lambda\gamma \tag{19}$$

$$-\frac{Ce^{-T/\tau_{LTP}}}{A} = \frac{1-\lambda}{\gamma}, \tag{20}$$

where $A, B, B'$ and $C$ are defined as in *Equations 14, 15, and 16*.

Hence, our plasticity rule is learning the Successor Representation through a TD($\lambda$) model with parameters:

$$\eta = -A \tag{21}$$

$$\gamma = \frac{A-C}{A}e^{-\frac{T}{\tau_{LTP}}} \tag{22}$$

$$\lambda = \frac{A}{A-C} \tag{23}$$

To ensure the learning rate $\eta$ is positive, one condition resulting from *Equation 21* is

$$A_{pre} > A_{LTP}\,N_{\text{pop}}\tau_{LTP}\,\tau_m\epsilon_0\left(\rho^{pre}\,(1-e^{-\theta/\tau_m})\frac{\theta-\tau_{LTP}(1-e^{-\theta/\tau_{LTP}})}{\theta} + \frac{1}{\tau_m+\tau_{LTP}}\right) \tag{24}$$

## Learning during normal behavior ($\theta >> \tau_{LTP}$)

During normal behavior, we assume the place-tuned currents are on larger timescales than the plasticity constants: $\theta, \omega >> \tau_{LTP}$. We can see from *Equations 14 and 16* that the factor $A$ grows linearly with $\theta$ while $C$ grows exponentially with $\theta$. From *Equation 23*, we then have

$$\lambda \to 0 \tag{25}$$

(See also *Figure 2—figure supplement 1*).

## Learning during replays ($\theta << \tau_{LTP}$)

### Assumptions

For the replay model we assume the place-tuned currents are impulses, which make the neurons emit exactly one spike at a given time. Specifically, we can make the duration of the place-tuned currents go to 0,

$$\theta, \omega \to 0 \tag{26}$$

while the intensity of the currents goes to infinity. For simplicity, we will take:

$$\rho^{pre}(\theta) = \frac{1}{\theta} \to \lim_{\theta \to 0}\rho^{pre} = \infty \qquad \rho^{bias}(\omega) = \frac{1}{\omega} \to \lim_{\omega \to 0}\rho^{bias} = \infty$$

Furthermore, we assume that the contribution of the postsynaptic currents due to the single presynaptic spikes is negligible in terms of driving plasticity, allowing us to set

$$\epsilon_0 \to 0$$

### Calculations of TD parameters

Given the assumptions above, we can see from *Equations 14 and 16* that:

$$A = -\eta_{STDP}A_{pre}$$

$$C = 0$$

For *Equation 15*, we can use the Taylor expansion for $e^{\frac{x}{\tau}}$ around $x = 0$, such that: $e^{\frac{x}{\tau}} \approx 1 + \frac{x}{\tau}$ :

$$\begin{aligned}B &= \eta_{STDP}A_{LTP}\tau_{LTP}^2\rho^{pre}\frac{\theta}{\tau_{LTP}}e^{-\frac{t^*}{\tau_{LTP}}}\frac{\omega}{\tau_{LTP}}\rho^{bias}\\&= \eta_{STDP}A_{LTP}e^{-\frac{t^*}{\tau_{LTP}}}\end{aligned}$$

Using *Equations 21, 22, 23 and 18*, we can calculate the parameters and constraints for the TD model:

$$\lambda = \frac{A}{A-C} = 1$$

$$\eta = -A = \eta_{STDP} A_{pre}$$

$$\gamma = \frac{A-C}{A} e^{-\frac{T}{\tau_{LTP}}} = e^{-\frac{T}{\tau_{LTP}}}$$

$$1 = -\frac{B}{A} = \frac{A_{LTP} e^{-\frac{t^*}{\tau_{LTP}}}}{A_{pre}}$$

(27)

As expected, the bootstrapping parameter $\lambda = 1$ (see also *Figure 2—figure supplement 1*).

## Alternative derivation of replay model

### Place cell activation during replays

We model a replay event as a precise temporal sequence of spikes. Since every neuron represents a state in the environment, a replay sequence reproduces a trajectory of states. We assume that, when the agent is in state $S = j$, the neurons representing state $j$ fire $n_{pre}$ spikes at some point in the time interval $t \in [0, \sigma]$, where the exact firing times are uniformly sampled. After a short delay, the CA1 neurons representing state $j$ fire $n_{post}$ spikes at a time uniformly sampled from the interval $[t^*, t^* + \sigma]$. The time between two consecutive state visits is $T$. The exact number of spikes in each replay event is random but small. Specifically, it is sampled from the set $\{0, 1, 2\}$ according to the probability vector

$$\mathbf{p} = (\frac{p_1}{2}, 1 - p_1, \frac{p_1}{2})$$

(28)

It is worth noting here that other implementations are possible but that we assume the average number of spikes in each state is 1, and that the average time between a presynaptic and a postsynaptic spike is $t^*$. The model could be further generalized for a higher number of average spikes per state.

### Plasticity update

We can consider again our learning rule, composed of a positive pre-post potentiation window and presynaptic weight-dependent depression (*Equation 9*). Let's consider the synapse $w_{ij}$, we can see that on average the total amount of depression will be determined by the number of times the state $j$ is visited in the trajectory replayed:

$$LTD = -A_{pre} \cdot w_{ij} N_j,$$

where $N_j$ is the number of times the state $j$ is visited. The amount of potentiation will be determined, instead, by the time difference between the postsynaptic and presynaptic firing times, which encode the distance between state $j$ and state $i$:

$$LTP = A_{LTP} \sum_k e^{-\frac{kT+t^*}{\tau_{LTP}}} n_k^{ij},$$

where $n_k^{ij}$ represents the number of times the agent visited state $i$ k steps after $j$. Combining the equations above we find that:

$$\Delta w_{ij} = \eta_{STDP} A_{LTP} \sum_k e^{-\frac{kT+t^*}{\tau_{LTP}}} n_k^{ij} - \eta_{STDP} A_{pre} \cdot w_{ij} N_j.$$

(29)

If we assume that the this value has converged to its stationary state, $\Delta w_{ij} = 0$;

$$w_{ij}^{\star} = \frac{A_{LTP}}{A_{pre}} e^{-\frac{t^*}{\tau_{LTP}}} \cdot \frac{\sum_k (e^{-\frac{T}{\tau_{LTP}}})^k n_k^{ij}}{N_j}$$

(30)

### Comparison with online Monte Carlo learning

Given the stable weight $w^*$ from *Equation 30*, we can impose that:

$$\frac{A_{LTP}}{A_{pre}} e^{-\frac{t^*}{\tau_{LTP}}} = 1 \qquad \text{and}$$

(31)

$$e^{-\frac{T}{\tau_{\mathrm{LTP}}}} = \gamma \tag{32}$$

we find that the stable weight is:

$$w_{ij}^{\star} = \frac{\sum_k \gamma^k n_k^{ij}}{N_j} \approx \mathbb{E}\left[\sum_k \gamma^k \mathbb{I}(S_k = i | S_0 = j)\right] = M(j, i) \tag{33}$$

which is the definition of the Successor Representation matrix (**Equation 4**). Indeed, $w_{ij}^{\star}$ is computing the sample mean of the discounted distance between states $i$ and $j$, which is equivalent to performing an every-state Monte Carlo or TD($\lambda$=1) update. Notably, from **Equation 29**, we have that the learning rate for the Monte Carlo update is given by:

$$\eta = \eta_{STDP} A_{\mathrm{LTP}} e^{-\frac{t^*}{\tau_{\mathrm{LTP}}}} = \eta_{STDP} A_{\mathrm{pre}} \tag{34}$$

## Simulation details for Figure 2

A linear track with four states is simulated. The policy of the agent in this simulation is to traverse the track from left to right, with one epoch consisting of starting in state 1 and ending in state 4. One simulation consists of 50 epochs, and we re-run the whole simulation ten times with different random seeds. Over these ten seeds, mean and standard deviation of the synaptic weights are recorded after every epoch.

Our neural network consists of two layers, each with a single neuron per state (as in **Figure 1**). Synaptic connections are made from each presynaptic neuron to all postsynaptic neurons, resulting in a 4-by-4 matrix which is initialized as the identity matrix. The plasticity rule and neuronal activations follow **Equations 9–12**.

The STDP parameters are listed in **Table 1**.

To obey **Equation 24**, we set $A_{\mathrm{pre}}$ equal to the right hand side augmented with 5.

For the behavioral case, we choose T=100ms, $\theta$=80ms, $\omega = T - \theta$, which correspond to TD($\lambda$) parameters $\lambda = 0.21$, $\gamma = 0.89$, $\eta = 0.12$.

In the replay case, we have a sequence of single spike per neuron (see **Figure 2b** and section 'Alternative derivation of replay model'). Following **Equation 27**, we choose $T = -\log(\gamma)\tau_{LTP} \approx$ 7ms, where $\gamma$ and $\tau_{LTP}$ are the same as in **Table 1**. We set $\theta = 2$ ms and $\sigma = 0.5$ ms. By setting the $\eta_{stdp} = \frac{\eta}{A_{LTP}\exp(\theta/\tau_{LTP})}$, the corresponding TD($\lambda$) parameters are $\lambda = 1$, $\gamma = 0.89$, $\eta = 0.12$ just as in the behavioral case.

More details on the place cell activation during replays in our model can be found in section 'Alternative derivation of replay model'. Using exactly one single spike per neuron with the above parameters would allow us to follow the TD(1) learning trajectories without any noise. For more biological realism, we choose $p_1 = 0.15$ in **Equation 28**, in order to achieve an equal amount of noise due to the random spiking as in the case of behavioral activity (see **Figure 4—figure supplement 2**).

**Table 1.** Parameters used for the spiking network.

| | |
|---|---|
| $\epsilon_0$ | 1 |
| $\rho_{\mathrm{pre}}$ | 0.1ms⁻¹ |
| $\tau_{\mathrm{m}}$ | 2ms |
| $N_{post}$ | 1 |
| $N_{pre_{tot}}$ | 1 |
| $N_{pre}$ | 1 |
| stepsize | 0.01ms |
| $\eta_{\mathrm{stdp}}$ | 0.003 |
| $\tau_{\mathrm{LTP}}$ | 60ms |
| $A_{LTP}$ | 1 ms⁻¹ |

## Simulation details for Figure 3

Using the same neural network and plasticity parameters as the behavioral learning in *Figure 2* (see previous section), we simulate the linear track in the following two situations:

- The third state has T=200ms instead of 100ms. All other parameters remain the same as in *Figure 2*. Results plotted in *Figure 3E*.
- The third state has $\rho_{\mathrm{pre}} = 0.2 \ \mathrm{ms}^{-1}$ instead of $0.1 \ \mathrm{ms}^{-1}$. All other parameters remain the same as in *Figure 2*. Results plotted in *Figure 3F*.

## Simulation details for Figure 4

A linear track with three states is simulated, and the agent has 50% probability to move left or right in each state (see *Figure 4A*). One epoch lasts until the agent reaches one of the *STOP* locations.

We then use the same neural network and plasticity parameters as used for *Figure 2*. We simulate three scenarios:

- Only replay-based learning during all epochs (no behavioral learning). This scenario corresponds to *MC STDP* in *Figure 4B* and to *Figure 4C*.
- Mixed learning using both behavior and replays. The probability for an epoch to be a replay is decaying over time following $\exp(-i/6)$, with $i$ the epoch number. This scenario corresponds to *Mix STDP* in *Figure 4B* and to *Figure 4E*.
- Only behavioral learning during all epochs (no replays). This scenario corresponds to *TD STDP* in *Figure 4B* and to *Figure 4D*.

## Simulation details for Figure 5

A linear track with 21 states is simulated. The SR is initialized as the identity matrix, and the reward vector (containing the reward at each state) is also initialized as the zero vector. We simulate the learning of the SR during behavior using the theoretical TD(0) updates and during replays using the theoretical TD(1) updates. The value of each state is then calculated as the matrix-vector product between the SR and the reward vector, resulting in an initial value of zero for each state.

The policy of the agent is a softmax policy (i.e. the probability to move to neighboring states is equal to the softmax of the values of those neighboring states). The first time the agent reaches the leftmost state of the track (state 1), the negative reward of –2 is revealed, mimicking the shock in the actual experiments, and the reward vector is updated accordingly for this state.

We now simulate two scenarios: in the first scenario, the agent always follows the softmax policy and no replays are triggered (see *Figure 5D*, left panel). In the second scenario, every time the agent enters the dark zone from the light zone (i.e. transitions from state 12 to state 11 in our simulation), a replay is triggered from that state until the leftmost state (state 1) (see *Figure 5D*, right panel). Both scenarios are simulated for 2000 state transitions. We then run these two scenarios 100 times and calculate mean and standard deviation of state occupancies (*Figure 5F*).

Finally, since the second scenario has more SR updates than the first scenario, we also simulate the first scenario for 4000 state transitions (*Figure 5—figure supplement 1*) and show how the observed behavior of *Figure 5* is unaffected by this.

# Additional information

### Funding

| Funder | Grant reference number | Author |
| --- | --- | --- |
| Wellcome Trust | 200790/Z/16/Z | Claudia Clopath |
| Engineering and Physical Sciences Research Council | EP/R035806/1 | Claudia Clopath |
| Simons Foundation | 564408 | Claudia Clopath |

| Funder | Grant reference number | Author |
|--------|------------------------|--------|

The funders had no role in study design, data collection and interpretation, or the decision to submit the work for publication. For the purpose of Open Access, the authors have applied a CC BY public copyright license to any Author Accepted Manuscript version arising from this submission.

## Author contributions

Jacopo Bono, Sara Zannone, Conceptualization, Data curation, Software, Formal analysis, Validation, Investigation, Visualization, Methodology, Writing - original draft; Victor Pedrosa, Visualization, Writing - original draft; Claudia Clopath, Conceptualization, Resources, Formal analysis, Supervision, Funding acquisition, Validation, Investigation, Visualization, Project administration, Writing - review and editing

## Author ORCIDs

Jacopo Bono (ORCID) http://orcid.org/0000-0001-9552-3151
Sara Zannone (ORCID) http://orcid.org/0000-0002-9526-7001
Claudia Clopath (ORCID) http://orcid.org/0000-0003-4507-8648

## Decision letter and Author response

Decision letter https://doi.org/10.7554/eLife.80671.sa1
Author response https://doi.org/10.7554/eLife.80671.sa2

# Additional files

## Supplementary files

• MDAR checklist

## Data availability

The current manuscript is a computational study, so no data have been generated for this manuscript. Modelling code is available on GitHub (https://github.com/jacopobono/learning_cognitive_maps_code, copy archived at swh:1:rev:d86b262545547353c7050bbc2d476c2f4a297989; *Jacopo, 2023*).

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

## Appendix 1

### Analytical derivations for the total weight change in the behavioural model

#### Presynaptic rate during state $j$

Whenever a state $S = j$ is entered, the presynaptic neurons encoding state $j$ start firing at a constant rate $\rho^{pre}$ for a time $\theta$, following a Poisson process with parameter $\rho_j^{pre}(t)$:

$$\rho_j^{pre}(t) = \begin{cases} \rho_{pre}, & \text{if } t \in [0, \theta) \\ 0, & \text{otherwise} \end{cases} \tag{35}$$

The other presynaptic neurons are silent.

#### Postsynaptic rate during state $j$

The average postsynaptic rate can be calculated as follows. The probability of a presynaptic spike between $t$ and $t + dt$ is equal to $\rho_j(t)dt$. The size of the presynaptic population encoding state $j$ is equal to $N_{pop}$ and each excitatory postsynaptic potential (EPSP) is modeled by an immediate jump with amplitude $\epsilon_0 w_{ij}$, followed by exponential decay of EPSP with time constant $\tau_m$.

Following **Equation 12** in the main paper, reproduced below,

$$\rho_i^{post}(t) = \sum_k^{N_{pop}} \sum_{t_k^f < t} w_{ij_k}(t)\kappa(t - t^f) + \rho_i^{bias}(t)$$

we find that the average postsynaptic potential at time $t$ is given by (assuming t=0 when entering the state $j$):

$$\bar{\rho}_i^{post}(t) = \int_0^t [N_{pop}\rho_j^{pre}(t')\epsilon_0 w_{ij}(t')e^{-\frac{t'}{\tau_m}} + \rho_i^{bias}(t')]dt' \tag{36}$$

We assume that $w_{ij}(t)$ changes slowly compared to the timescale $\theta$ allowing us to consider the weight constant during that time. We can then approximate the average postsynaptic rate as:

$$\bar{\rho}_i^{post}(t) = \begin{cases} N_{pop}\rho^{pre}\epsilon_0 w_{ij}\tau_m(1 - e^{-\frac{t}{\tau_m}}), & \text{if } 0 \leq t < \theta \\ \rho^{bias}\delta_{ij}, & \text{if } t^* \leq t < t^* + \omega \\ 0, & \text{otherwise} \end{cases} \tag{37}$$

If $t^* < \theta$, both the first and the second term will contribute to the postsynaptic rate in the time between $t^*$ and $\theta$.

#### LTP trace during state $j$

Given **Equation 9** in the main paper, reproduced below,

$$\tau_{LTP}\frac{dTr_{LTP}^j(t)}{dt} = -Tr_{LTP}^j(t) + \sum_j \delta(t - t^j)$$

and combined with **Equation 35**, we can calculate the evolution of the LTP trace for neuron $j$ during state $j$:

$$
\begin{aligned}
Tr_{LTP}^j(t) &= \begin{cases} \rho^{pre}\int_0^t e^{-\frac{t'}{\tau_{LTP}}}dt', & \text{if } 0 \leq t < \theta \\ \rho^{pre}\tau_{LTP}(1 - e^{-\frac{\theta}{\tau_{LTP}}})e^{-\frac{t-\theta}{\tau_{LTP}}}, & \text{if } t \geq \theta \end{cases} \\
&= \begin{cases} \rho^{pre}\tau_{LTP}(1 - e^{-\frac{t}{\tau_{LTP}}}), & \text{if } 0 \leq t < \theta \\ \rho^{pre}\tau_{LTP}(e^{\frac{\theta}{\tau_{LTP}}} - 1)e^{-\frac{t}{\tau_{LTP}}}, & \text{if } t \geq \theta \end{cases}
\end{aligned} \tag{38}
$$

For $0 \leq t < \theta$, the presynaptic neuron $j$ is active and therefore the trace builds up with the presynaptic spikes, for $t \geq \theta$, the trace decays exponentially with time constant $\tau_{LTP}$.

## Total amount of LTP during state $j$

Following (**Kempter et al., 1999**), first we calculate the amount of LTP without taking into account spike-to-spike correlation:

The probability for a postsynaptic spike between $t$ and $t + dt$ is $\bar{\rho}_i^{post}(t)\,dt$. The amount of LTP due to a single spike at time $t$ is $A_{LTP}\,Tr_{LTP}^j(t)$. Hence, combining **Equations 37 and 38**, the total amount of LTP during a state (i.e. between time 0 and $T$) becomes:

$$LTP_{\text{non-causal}} \quad = A_{LTP}\int_0^T \bar{\rho}_i^{post}(t)\,Tr_{LTP}^j(t)\,dt \tag{39}$$

$$
\begin{aligned}
&= A_{LTP}N_{pop}\rho^{pre}\epsilon_0 w_{ij}\tau_m(1 - e^{-\frac{\theta}{\tau_m}})\rho^{pre}\tau_{LTP}\int_0^\theta(1 - e^{-\frac{t'}{\tau_{LTP}}})dt' \\
&\quad + A_{LTP}\rho^{bias}\,\rho^{pre}\tau_{LTP}(e^{\frac{\theta}{\tau_{LTP}}} - 1)e^{-\frac{t^*}{\tau_{LTP}}}\int_{t^*}^{t^*+\omega} e^{-\frac{t'}{\tau_{LTP}}}\,dt' \\
&= A_{LTP}\,w_{ij}\,N_{pop}\epsilon_0(\rho^{pre})^2\,\tau_{LTP}\,\tau_m(1 - e^{-\frac{\theta}{\tau_m}})\left[\theta - \tau_{LTP}(1 - e^{-\frac{\theta}{\tau_{LTP}}})\right] \\
&\quad + A_{LTP}\rho^{bias}\,\rho^{pre}\tau_{LTP}^2(e^{\frac{\theta}{\tau_{LTP}}} - 1)e^{-\frac{t^*}{\tau_{LTP}}}[1 - e^{-\frac{\omega}{\tau_{LTP}}}]
\end{aligned}
\tag{40}
$$

Following (**Kempter et al., 1999**), the amount of LTP due to the causal part (each presynaptic spike temporarily increase the probability of a postsynaptic spike) is given by:

$$LTP_{\text{causal}} = A_{LTP}\theta\rho^{pre}\epsilon_0 w_{ij}\frac{\tau_m\tau_{LTP}}{\tau_m+\tau_{LTP}} \tag{41}$$

Combining equations for the non-causal 40 and causal 41 parts, we get the *total* amount of LTP during a state (assuming $\tau_m << \tau_{LTP}$):

$$
\begin{aligned}
LTP \quad &= A_{LTP}\,w_{ij}\,N_{pop}\epsilon_0(\rho^{pre})^2\,\tau_{LTP}\,\tau_m(1 - e^{-\frac{\theta}{\tau_m}})\left[\theta - \tau_{LTP}(1 - e^{-\frac{\theta}{\tau_{LTP}}})\right] \\
&\quad + A_{LTP}\rho^{bias}\,\rho^{pre}\tau_{LTP}^2(e^{\frac{\theta}{\tau_{LTP}}} - 1)e^{-\frac{t^*}{\tau_{LTP}}}[1 - e^{-\frac{\omega}{\tau_{LTP}}}] \\
&\quad + A_{LTP}\theta\rho^{pre}\frac{\tau_m\tau_{LTP}}{\tau_m+\tau_{LTP}}\epsilon_0 w_{ij}
\end{aligned}
\tag{42}
$$

## Total amount of LTD during state $j$

There is a weight-dependent depression for each presynaptic spike, hence the amount of LTD during a state is given by:

$$LTD = -A_{pre}\rho^{pre}\theta w_{ij} \tag{43}$$

## Total plasticity during state $j$

Combining **Equations 42 and 43**, we can calculate the total amount of plasticity during the time the agent spends in the current state $j$:

$$\Delta^0 w_{ij} = A\,w_{ij} + B\delta_{ij} \tag{44}$$

with

$$
\begin{aligned}
A \quad &= \eta_{STDP}A_{LTP}\,N_{pop}\epsilon_0(\rho^{pre})^2\,\tau_{LTP}\,\tau_m(1 - e^{-\theta/\tau_m})\left[\theta - \tau_{LTP}(1 - e^{-\theta/\tau_{LTP}})\right] \\
&\quad + \eta_{STDP}A_{LTP}\theta\rho^{pre}\frac{\tau_m\tau_{LTP}}{\tau_m+\tau_{LTP}}\epsilon_0 - \eta_{STDP}A_{pre}\,\rho^{pre}\theta
\end{aligned}
\tag{45}
$$

and

$$B = \eta_{STDP}A_{LTP}\,\rho^{pre}\tau_{LTP}^2(e^{\frac{\theta}{\tau_{LTP}}} - 1)e^{-\frac{t^*}{\tau_{LTP}}}[1 - e^{-\frac{\omega}{\tau_{LTP}}}]\rho^{bias} \tag{46}$$

## Plasticity due to states transitioning

Once the agent leaves state $j$, the decaying LTP trace can still cause potentiation due to the activity in the following states, $j + n$, with $n = 1, 2, \ldots$. Given that the agent spends a time $T$ in each state, we

find that the agent visits state $j + n$ during time $t \in [nT, nT + T)$. We will now calculate the contribution to plasticity due to these state transitions.

## Postsynaptic rate during the new state $j + n$

During state $j + n$, the activity of the postsynaptic neurons is driven by the presynaptic neurons coding for $j + n$, and the bias current. We can thus generalize *Equation 37* and find that the average postsynaptic rate $\bar{\rho}_i^{post}$ during state $j + n$ is:

$$\bar{\rho}_i^{post}(t) = \begin{cases} N_{pop}\rho^{pre}\epsilon_0 w_{ij+n}\tau_m(1 - e^{-\frac{\theta}{\tau_m}}), & \text{if } nT \leq t < nT + \theta \\ \rho^{bias}\delta_{ij+n}, & \text{if } nT + t^* \leq t < nT + t^* + \omega \\ 0, & \text{otherwise} \end{cases} \tag{47}$$

## LTP trace from state $j$, during the new state $j + n$

Following *Equation 38*, we find that the amplitude of the LTP trace from state $j$ during state $j + n$ is:

$$\begin{aligned} Tr_{LTP}^j(nT + t') &= \rho^{pre}\tau_{LTP}(e^{\frac{\theta}{\tau_{LTP}}} - 1)e^{-\frac{(t' + nT)}{\tau_{LTP}}} \\ &= \rho^{pre}\tau_{LTP}(e^{\frac{\theta}{\tau_{LTP}}} - 1)(e^{-\frac{T}{\tau_{LTP}}})^n e^{-\frac{t'}{\tau_{LTP}}} \end{aligned} \tag{48}$$

with $0 < t' < T$.

## LTP due to state transitioning

We can then calculate the amount of LTP between the presynaptic neuron $j$ and the postsynaptic neuron $i$, when the agent is in state $j + n$. We refer to *Equation 39* and find:

$$\begin{aligned} LTP_{\text{switch}} &= A_{LTP}\int_{nT}^{nT+T}\bar{\rho}_i^{post}(t') Tr_{LTP}^j(t')\, dt' \\ &= A_{LTP}N_{pop}\rho^{pre}\epsilon_0 w_{ij+n}\tau_m(1 - e^{-\frac{\theta}{\tau_m}})\rho^{pre}\tau_{LTP}(e^{\frac{\theta}{\tau_{LTP}}} - 1)(e^{-\frac{T}{\tau_{LTP}}})^n \int_{nT}^{nT+\theta} e^{-\frac{t'}{\tau_{LTP}}}\, dt' \\ &\quad + A_{LTP}\rho^{bias}\delta_{ij+n}\rho^{pre}\tau_{LTP}(e^{\frac{\theta}{\tau_{LTP}}} - 1)e^{-\frac{t^*}{\tau_{LTP}}}(e^{-\frac{T}{\tau_{LTP}}})^n \int_{nT+t^*}^{nT+t^*+\omega} e^{-\frac{t'}{\tau_{LTP}}}\, dt' \\ &= A_{LTP}N_{pop}\rho^{pre}\epsilon_0 w_{ij+n}\tau_m(1 - e^{-\frac{\theta}{\tau_m}})\rho^{pre}\tau_{LTP}(e^{\frac{\theta}{\tau_{LTP}}} - 1)(e^{-\frac{T}{\tau_{LTP}}})^n \tau_{LTP}(1 - e^{-\frac{\theta}{\tau_{LTP}}}) \\ &\quad + A_{LTP}\rho^{bias}\delta_{ij+n}\rho^{pre}\tau_{LTP}(e^{\frac{\theta}{\tau_{LTP}}} - 1)e^{-\frac{t^*}{\tau_{LTP}}}(e^{-\frac{T}{\tau_{LTP}}})^n \tau_{LTP}(1 - e^{-\frac{\omega}{\tau_{LTP}}}) \end{aligned}$$

The amount of plasticity in state $j + n$ when starting from state $j$ is thus:

$$\Delta^n w_{ij} = C(e^{-\frac{T}{\tau_{LTP}}})^n w_{ij+n} + B(e^{-\frac{T}{\tau_{LTP}}})^n \delta_{ij+n} \tag{49}$$

where

$$C = \eta_{STDP}A_{LTP}N_{pop}\rho^{pre}\epsilon_0\tau_m(1 - e^{-\frac{\theta}{\tau_m}})\rho^{pre}\tau_{LTP}(e^{\frac{\theta}{\tau_{LTP}}} - 1)\tau_{LTP}(1 - e^{-\frac{\theta}{\tau_{LTP}}}) \tag{50}$$

$$B = \eta_{STDP}A_{LTP}\rho^{pre}\tau_{LTP}^2(e^{\frac{\theta}{\tau_{LTP}}} - 1)e^{-\frac{t^*}{\tau_{LTP}}}(1 - e^{-\frac{\omega}{\tau_{LTP}}})\rho^{bias} \tag{51}$$

It is worth noting that the parameter B derived here is the same as *Equation 46*.

## Summary: total STDP update

If we combine together *Equations 44 and 49*, we have that the total weight change for the synapse $w_{ij}$ is given by:

$$w_{ij} = \Delta^0 w_{ij} + \sum_{n=1}^{N}\Delta^n w_{ij} = A\, w_{ij} + \sum_{n=0}^{N}[B(e^{-T/\tau_{LTP}})^n \delta_{ij+n} + C(e^{-T/\tau_{LTP}})^{n+1} w_{i,j+n+1}] \tag{52}$$

where $N$ is the number of states until the end of the trajectory and A, B, C are as defined in *Equations 45, 46 and 50* respectively.

## Analytical calculations for hyperbolic discounting

From **Equation 22** in the main paper, we have that, in the behavioural model $\gamma = (1 - \frac{C}{A})e^{-\frac{T}{\tau_{LTP}}}$. Here, we will derive an approximation to this value.

If we assume that $\theta \gg \tau_m, \tau_{LTP}$, we can approximate $A$ and $C$ as:

$$
\begin{aligned}
\tilde{A} &= \eta_{STDP}A_{LTP}\,N_{\text{pop}}\epsilon_0(\rho^{pre})^2\,\tau_{LTP}\,\tau_m(\theta - \tau_{LTP}) \\
&\quad + \eta_{STDP}A_{LTP}\theta\rho^{pre}\tfrac{\tau_m\tau_{LTP}}{\tau_m+\tau_{LTP}}\epsilon_0 - \eta_{STDP}A_{pre}\,\rho^{pre}\theta \\
&= \eta_{STDP}\rho^{pre}(a\theta + b),
\end{aligned} \tag{53}
$$

$$
\text{with} \quad
\begin{aligned}
a &= A_{LTP}\,\epsilon_0\tau_{LTP}\,\tau_m(N_{\text{pop}}\rho^{pre} + \tfrac{1}{\tau_m+\tau_{LTP}}) - A_{pre} \\
b &= -A_{LTP}\,N_{\text{pop}}\epsilon_0\rho^{pre}\tau_{LTP}^2\,\tau_m
\end{aligned} \tag{54}
$$

$$
\begin{aligned}
\tilde{C} &= \eta_{STDP}A_{LTP}N_{pop}(\rho^{pre})^2\epsilon_0\tau_m\tau_{LTP}^2 e^{\frac{\theta}{\tau_{LTP}}} \\
&= -\eta_{STDP}\rho^{pre}e^{\frac{\theta}{\tau_{LTP}}}\cdot b
\end{aligned} \tag{55}
$$

If we define $\psi$ such that $\theta + \psi = T$, we can rewrite and approximate the discount parameter as:

$$
\begin{aligned}
\gamma &= (1 - \frac{C}{A})e^{-\frac{\theta+\psi}{\tau_{LTP}}} \approx -\frac{\tilde{C}}{\tilde{A}}e^{-\frac{\theta}{\tau_{LTP}}}e^{-\frac{\psi}{\tau_{LTP}}} \\
&= \frac{be^{-\frac{\psi}{\tau_{LTP}}}}{a\theta+b} = \frac{1}{1+\frac{a}{b}\theta}\cdot e^{-\frac{\psi}{\tau_{LTP}}}
\end{aligned} \tag{56}
$$

From **Equation 56**, we can see that the discount $\gamma$ follows a hyperbolic function if we increase the duration of the presynaptic current $\theta$. If, instead, we vary $\psi$, the discount becomes exponential (**Figure 2—figure supplement 1a and b**).

Notice that this analysis extends to the replay model. Following what was done after **Equation 26**, we can connect the behavioural model with the replay model by making $\theta, \epsilon_0 \to 0$, which implies $\psi \to T$. From **Equation 56** we find that:

$$
\lim_{\theta,\epsilon_0 \to 0} \gamma = e^{-\frac{T}{\tau_{LTP}}},
$$

which is exactly the definition of $\gamma$ in the replay model (**Equations 27** in Materials and methods). For replays, the discount is therefore strictly exponential.

Furthermore, using the same calculations and **Equations 21 and 19** in the main paper, we can find approximated values for the other parameters too (**Figure 2—figure supplement 1c and d**).

$$
\begin{aligned}
\eta &= -A \approx -\eta_{STDP}\rho^{pre}(a\theta + b) \\
\lambda &= \frac{e^{-T/\tau_{LTP}}}{\gamma} \approx (1+\tfrac{a}{b}\theta)e^{\frac{\psi}{\tau_{LTP}}}e^{-\frac{\psi+\theta}{\tau_{LTP}}} = (1+\tfrac{a}{b}\theta)e^{-\frac{\theta}{\tau_{LTP}}}
\end{aligned}
$$

