## [Editor Report]

This is an important article that leverages a spiking network model of the hippocampal circuit to show how spike-time-dependent plasticity can implement predictive reinforcement learning and form a predictive map of the environment. The authors provide a convincing and solid framework for understanding the prediction based learning rules that may be employed by the hippocampus to optimize an animal's behavior. This paper will be of interest to theoretical and experimental neuroscientists working on learning and memory as it provides new ways to connect computational models to experimental data that has yet to be fully explored from a reinforcement learning perspective.

---

## [Decision Letter]

**Decision letter after peer review:**

Thank you for submitting your article "Learning predictive cognitive maps with spiking neurons during behaviour and replays" for consideration by *eLife*. Your article has been reviewed by 3 peer reviewers, and the evaluation has been overseen by a Reviewing Editor and Laura Colgin as the Senior Editor. The following individual involved in review of your submission has agreed to reveal their identity: Michael E. Hasselmo (Reviewer #2).

Essential revisions:

Based on the comments from all reviewers, I'd recommend focusing your revisions on the primary topic of improving the link between their model and experimental data. Specifically, this would include: Consideration (or modeling) of how the model would extend to 2D, discussion on how the neural activity would be used to perform computations, the limitations of RL as it relates to interpreting experimental data, and to more appropriately frame the work in the context of experimental studies (all reviewers had detailed suggestions for how to do this with text changes). I've provided highlights from the three reviewers below that apply to these concerns but note that reviewer 2 also provided a number of specific suggestions related to text/reference changes. All reviewer comments are also included at the bottom of this message.

Reviewer 1:

– The successor representation is learned at the level of synaptic weights between the two layers. It is not clear how it is read out into neural activity and exploited to perform actual computations, as both layers are assumed to be strongly driven by external inputs. This is a major limitation of this work.

– One of the results is that STDP at the timescale of milliseconds can lead to learning over behavioral timescales of seconds. This result seems related to Drew and Abbott PNAS 2006. In that work, the mapping between learning on micro and macro timescales in fact relied on precise tuning of plasticity parameters. It is not clear to which extent similar limitations apply here, and what is the precise relation with Drew and Abbott.

– Most of the results are presented at a formal, descriptive level relating plasticity to reinforcement learning algorithms. The provided examples are quite limited and focus on a simplified setting, a linear track. It would be important to see that the results extend to two-dimensional environments, and to show how the successor representation is actually used (see first comment).

– The main text does not explain clearly how replays are implemented.

Reviewer 2:

I think the authors of this article need to be clear about the shortcomings of RL. They should devote some space in the discussion to noting neuroscience data that has not been addressed yet. They could note that most components of their RL framework are still implemented as algorithms rather than neural models. They could note that most RL models usually don't have neurons of any kind in them and that their own model only uses neurons to represent state and successor representations, without representing actions or action selection processes. They could note that the agents in most RL models commonly learn about barriers by needing to bang into the barrier in every location, rather than learning to look at it from a distance. The ultimate goal of research such as this should be to link cellular level neurophysiological data to experimental data on behavior. To the extent possible, they should focus on how they link neurophysiological data at the cellular level to spatial behavior and the unit responses of place cells in behaving animals, rather than basing the validity of their work on the assumption that the successor representation is correct.

Reviewer 3:

1. Could the authors elaborate more on the connection between the biological replays that are observed in a different context in the brain and the replays implemented in their model? Within the modeling context, when are replays induced upon learning in a novel environment, and what is the influence of replays when/if they are generated upon revisiting the previously seen/navigated environment?

2. The model is composed of CA1 and CA3, what are the roles of the other hippocampal subregions in learning predictive maps? From the reported results, it looks like it may be possible that prediction-based learning can be successfully achieved simply via the CA1-CA3 circuit. Are there studies (e.g., lesioned) that show this causal relationship to behavior? Along this line, what are the potential limitations of the proposed framework in understanding the circuit computation adopted by the hippocampus?

3. Do the authors believe that the plasticity rules/computational principles observed within the 2-layer model are specific to the CA1-CA3 circuit? Can these rules be potentially employed elsewhere within the medial temporal lobe or sensory areas? What are the model parameters used that could suggest that the observed results are specific to hippocampus-based predictive learning?

4. The analytical illustration linking the proposed model with reinforcement learning is well executed. However, in practice, the actual implementation of reinforcement learning within the model is unclear. Given the sample task provided where animals are navigating a simple environment, how can one make use of value-based learning to enhance behavior? Explicit discussion on the extent to which reinforcement learning is related to the actual computation potentially needed to navigate sensory environments (both learned and novel) would be really helpful in understanding the link between the model to reinforcement learning.

5. Subplots both within and across figures seem to be of very different text formatting and sizing (such as panel F in Figure 4 and Figure 5). Please reformat them accordingly.

*Reviewer #2 (Recommendations for the authors):*

Important: Note that the page numbers refer to the page in the PDF, which is their own page number-1 (due to *eLife* adding a header page).

Page 3 – "smoothly… and anything in between" – this is overstated and should be removed.

Page 3 – "don't need to discretize time…". Here and elsewhere there should be citations to the work of Doya, NeurIPS 1995, Neural Comp 2000 on the modeling of continuous time in RL.

Page 3 – "using replays" – It is very narrowminded to assume that all replay do is set up successor representations. They could also be involved in model-based planning of behavior as suggested in the work of Johnson and Redish, 2012; Pfeiffer and Foster, 2018; Kay et al. 2021 and modeled in Hasselmo and Eichenbaum, 2005; Erdem and Hasselmo, 2012 and Fenton and Kubie, 2012.

Page 3 – They assume that STDP can occur during replay, but evidence for STDP during replay is unclear. McNaughton's lab showed that LTP is less likely to be induced during the modulatory states during which sharp-wave ripple replay events occur. They should look for citations that have actually shown LTP induction during the replay state.

Page 4 – Marr's three levels – They should remove this discussion about Marr's three levels as I think the implementation level is relatively sparse and the behavioral level is also relatively sparse.

Page 4 – "The hippocampus has long been thought" – It's astounding that the introduction only cites two experimental papers (O'Keefe and Dostrovsky, 1971; Mehta et al. 2000) and then the start of the Results section makes a statement like this and only cites Stachenfeld et al. 2017 as if it were an experimental paper. There are numerous original research papers that should be cited for the role of hippocampus in behavior. They should cite at least five or six so that the reader doesn't get the impression all o this work started with the holy paper by Stachenfeld et al. 2017. For example, they could again cite O'Keefe and Nadel, 1978 for the very comprehensive review of the literature up to that time, plus the seminal work of Morris et al. 1982 in Morris water maze and Olton, 1979 in 8-arm radial maze and perhaps some of the work by Aggleton and by Eichenbaum on spatial alternation.

Page 5 – The description of successor representations is very one dimensional. They should mention how it can be expanded to two dimensions.

Page 5 – "Usually attributed to model based…". They cannot just talk about SR being model free. Since this section was supposed to be for neuroscientists, they need to clearly explain the distinction between model free and model-based RL, and describe why successor representations are not just model-based RL, but instead provide a look-up table of predictive state that does NOT involve model-based planning of behavior. The blog of Vitay gives a much better overview that compares model-free, model-based and successor representations:

https://julien-vitay.net/post/successor_representations/ – This needs more than just a citation – there should be a clear description of model-based and model-free RL in contrast to SR, and Vitay is an example of that.

Page 5 – Related to this issue – they need to repeatedly address the fact that Successor representations are just an hypothesis contrast with model-based behavior, and repeatedly throughout the paper discuss that model-based behavior could still be the correct accounting for all of the data that they address.

Page 5 – "similar to (Mehta et al. 2000)" – Learning in the CA3-CA1 network has been modeled like this in many previous models that should be cited here including McNaughton and Morris, 1987; Hasselmo and Schnell, 1994; Treves and Rolls, 1994; Mehta et al. 2000; Hasselmo, Bodelon and Wyble, 2002.

Page 6 – Figure 1d looks like the net outcome of the learning rule in this example is long-term depression. Is that intended? Given the time interval between pre and post, it looks like it ought to be potentiation in the example.

Page 7 – They should address the problem of previously existing weights in the CA3 to CA1 connections. For example, what if there are pre-existing weights that are strong enough to cause post-synaptic spiking in CA1 independent of any entorhinal input? How do they avoid the strengthening of such connections? (i.e. the problem of prior weights driving undesired learning on CA3-CA1 synapses is addressed in Hasselmo and Schnell, 1994; Hasselmo, Bodelon and Wyble, 2002, which should be cited).

Page 7 – "Elegantly combines rate and temporal" – This is overstated. The possible temporal codes in the hippocampus include many possible representations beyond just one step prediction. They need to specify that this combines one type of possible temporal code. I also recommend removing the term "elegant" from the paper. Let someone else call your work elegant.

Page 7 – "replays for learning" – as noted above in experiments LTP has not been shown to be induced during the time periods of replay – sharp-wave ripple replay events seem to be associated with lower cholinergic tone (Buzsaki et al. 1983; VandeCasteele et al. 2014) whereas LTP is stronger when Ach levels are higher (Patil et al. J. Neurophysiol. 1998). This is not an all-or-none difference, but it should be addressed.

Page 7 – "equivalent to TD…". Should this say "equivalent to TD(0)"?

Page 8 – "Bootstrapping means that a function is updated using current estimates of the same function…". This is a confusing and vague description of bootstrapping. They should try to give a clearer definition for neuroscientists (or reduce their reference to this).

Page 9 – Figure 2 – Do TD λ and TD zero really give equivalent weight outputs?

Page 8 – "that are behaviorally far apart" – I don't understand how this occurs.

page 10 – "dependency of synaptic weights on each other as discussed above." This was not made sufficiently clear either here or above.

Page 10 – "dependency of synaptic weights on each other" This also suggests a problem of stability if the weights can start to drive their own learning and cause instability – how is this prevented?

Page 10 – "average of the discounted state occupancies" – this would be uniform without discounting but what is the biological mechanism for the discounting that is used here?

Page 10 – "due to the bootstrapping" – again this is unclear – can be improved by giving a better definition of bootstrapping and possibly by referring to specific equation numbers.

Page 11 – "exponential dependence" what is the neural mechanism for this?

Page 11 – "Ainsley" is not a real citation in the bibliography. Should fix and also provide a clearer definition (or equation) for hyperbolic.

Page 11 – "elegantly combines two types of discounting" – how is useful? Also, let other people call your work elegant.

Page 11 – how does discounting depend on both firing rate and STDP -- should provide some explanation or at least refer to where this is shown in the equations.

page 13 – "Cheng and Frank" – this is a good citation, but they could add more here on timing of replay events.

Page 15 – This whole section on the shock experiment starts with the assumption of a successor representation. As noted above, they need to explicitly discuss the important alternate hypothesis that the neural activity reflects model-based planning that guides the behavior in the task (and could perhaps better account for the peak of occupancy at the border of light and dark).

Page 16 – "mental imagination" – rather than using it for modifying SR, why couldn't mental imagination just be used for model-based behavior?

Page 17 – "spiking" – again, if they are going to refer to their model as a "spiking" model, they need to add some plots showing spiking activity.

*Reviewer #3 (Recommendations for the authors):*

I found the proposed modeling framework to be very exciting and of potential interest to not only computational neuroscientists but also to readers who are interested in neural mechanisms underlying learning in general. The manuscript is well-written and includes a detailed description and rationale of the model setups as well as the findings and their relevance to biological findings. That said, I have a few comments that I hope the authors could help address:

1. Could the authors elaborate more on the connection between the biological replays that are observed in a different context in the brain and the replays implemented in their model? Within the modeling context, when are replays induced upon learning in a novel environment, and what is the influence of replays when/if they are generated upon revisiting the previously seen/navigated environment?

2. The model is composed of CA1 and CA3, what are the roles of the other hippocampal subregions in learning predictive maps? From the reported results, it looks like it may be possible that prediction-based learning can be successfully achieved simply via the CA1-CA3 circuit. Are there studies (e.g., lesioned) that show this causal relationship to behavior? Along this line, what are the potential limitations of the proposed framework in understanding the circuit computation adopted by the hippocampus?

3. Do the authors believe that the plasticity rules/computational principles observed within the 2-layer model are specific to the CA1-CA3 circuit? Can these rules be potentially employed elsewhere within the medial temporal lobe or sensory areas? What are the model parameters used that could suggest that the observed results are specific to hippocampus-based predictive learning?

4. The analytical illustration linking the proposed model with reinforcement learning is well executed. However, in practice, the actual implementation of reinforcement learning within the model is unclear. Given the sample task provided where animals are navigating a simple environment, how can one make use of value-based learning to enhance behavior? Explicit discussion on the extent to which reinforcement learning is related to the actual computation potentially needed to navigate sensory environments (both learned and novel) would be really helpful in understanding the link between the model to reinforcement learning.

5. Subplots both within and across figures seem to be of very different text formatting and sizing (such as panel F in Figure 4 and Figure 5). Please reformat them accordingly.

[Editors’ note: further revisions were suggested prior to acceptance, as described below.]

Thank you for resubmitting your work entitled "Learning predictive cognitive maps with spiking neurons during behaviour and replays" for further consideration by *eLife*. Your revised article has been evaluated by Laura Colgin (Senior Editor) and a Reviewing Editor.

The manuscript has been improved but there are some remaining issues that need to be addressed, as outlined below:

Reviewer 1 makes two text suggestions that I believe would clarify the findings. There remains some lack of clarification around (1) how the second layer of the model mixes the successor representation with a representation of the current state itself and (2) justification for the difference in duration of the external inputs to the two layers. Reviewer 1 also suggests an additional figure, but I leave the decision to add (or not) this to the authors. Details regarding the requested clarifications are below.

*Reviewer #1 (Recommendations for the authors):*

The revised article has only partly resolved my confusion.

My main issue was the following: in the proposed feed-forward model, the synaptic weights between the two layers learn the entries of the successor matrix. If external inputs were fed only to the first layer, the second layer would directly read out the successor representation (this is suggested in Figure 1 E-F, but not explicitly mentioned in the text as far as I can tell). Instead, in the model, both layers are driven by external inputs representing the current state. This is crucial for learning, but it implies that the activity of the second layer mixes the successor representation with a representation of the current state itself. Learning and readout, therefore, seem antagonistic. It would be worth explaining this fact in the main text.

In their reply, the authors clarify that the external inputs drive the activity of the second layer only for a limited time (20%). As far as I can tell, in the text, this is mentioned explicitly only in the legend of Figure 5-S2. That seems to imply that there is a large difference in the duration of the external inputs to the two layers. How can that be justified?

More importantly, it seems that varying the value of the delay should lead to a tradeoff between the accuracy of learning and the accuracy of the subsequent readout in the second layer. Is that the case? It would be useful to have a figure where the delay is varied.

---

## [Author Response]

Reviewer 1:– The successor representation is learned at the level of synaptic weights between the two layers. It is not clear how it is read out into neural activity and exploited to perform actual computations, as both layers are assumed to be strongly driven by external inputs. This is a major limitation of this work.

We thank the reviewer for this important remark. Since we modelled our neurons to integrate the synaptic EPSPs and generated spikes using an inhomogeneous Poisson process based on the depolarization, the firing rate is proportional to the total synaptic weights. Therefore, the successor representation can be read out simply by a downstream neuron. Moreover, since the value of a state is defined by the inner product between the successor matrix and the reward vector, it is sufficient for the synaptic weights to the downstream neuron to learn the reward vector, and the downstream neuron will then encode the state value in its firing rate. We performed additional simulations and have added a supplementary figure (Figure 5—figure supplement S2) showing this setup.

We also want to note that the external inputs driving the activity affect the firing rate of the value neuron only during a limited amount of time (20% of the time in each state in our simulation of Figure 5—figure supplement S2). Moreover, this effect changes the estimate of the value quantitatively, but not qualitatively, i.e. the ranking of the states by value is not affected.

Practically, using the parameters in our simulation, one could either read out the correct estimate of the value during the first 80% of the time in a state, learn a correction to this perturbation in the weights to the value neuron, or simply use a policy that is based on the ranking of the values instead of the actual firing rate.

Besides the supplementary figure (Figure 5—figure supplement S2), we added the following paragraph in the Discussion section:

“Since we modelled our neurons to integrate the synaptic EPSPs and generate spikes using an inhomogeneous Poisson process based on the depolarization, the firing rate is proportional to the total synaptic weights. Therefore, the successor representation can be read out simply by a downstream neuron. Moreover, since the value of a state is defined by the inner product between the successor matrix and the reward vector, it is sufficient for the synaptic weights to the downstream neuron to learn the reward vector, and the downstream neuron will then encode the state value in its firing rate (see Figure 5—figure supplement S2). While the neuron model used is simple, it will be interesting future work to study analogous models with non-linear neurons.”

– One of the results is that STDP at the timescale of milliseconds can lead to learning over behavioral timescales of seconds. This result seems related to Drew and Abbott PNAS 2006. In that work, the mapping between learning on micro and macro timescales in fact relied on precise tuning of plasticity parameters. It is not clear to which extent similar limitations apply here, and what is the precise relation with Drew and Abbott.

We thank the reviewer for pointing us to the interesting work by Drew and Abbott. We added the following paragraph in the Discussion section:

“Learning on behavioural timescales using STDP was also investigated in Drew 2006. The main difference between Drew 2006 and our work, is that the former relies on overlapping neural activity between the pre- and post-synaptic neurons from the start, while in our case no such overlap is required. In other words, our setup allows us to learn connections between a presynaptic neuron and a postsynaptic neuron whose activities are separated by behavioural timescales initially. For this to be possible, there are two requirements: (1) the task needs to be repeated many times and (2) a chain of neurons are consecutively activated between the aforementioned presynaptic and postsynaptic neuron. Due to this chain of neurons, over time the activity of the postsynaptic neuron will start earlier, eventually overlapping with the presynaptic neuron.”

– Most of the results are presented at a formal, descriptive level relating plasticity to reinforcement learning algorithms. The provided examples are quite limited and focus on a simplified setting, a linear track. It would be important to see that the results extend to two-dimensional environments, and to show how the successor representation is actually used (see first comment).

We thank the reviewer for the feedback, and have now included two new supplementary figures. Figure 5—figure supplement S2 shows the usefulness of our setup when learning the state values, as discussed above. Figure 1—figure supplement S1shows a simulation in a 2D environment. In fact, due to the exact link with TD(λ), our setup is general for any type of task, in any dimension, where states are visited and which may not need to be a navigation task.

We have added the following sentences in the second to last paragraph of section 2.2:

Moreover, due to the equivalence with TD(λ), our setup is general for any type of task where discrete states are visited, in any dimension, and which may not need to be a navigation task (see e.g. Figure 2d for a 2D environment).

– The main text does not explain clearly how replays are implemented.

We thank the reviewer for pointing out this issue. We have now updated the methods section 4.3.6 to be more clear about the implementation of the replays. We have also updated the description of replay generation in the methods of Figure 2 as follows:

“More details on the place cell activation during replays in our model can be found in section 4.3.6. Using exactly one single spike per neuron with the above parameters would allow us to follow the TD(1) learning trajectories without any noise. For more biological realism, we choose p1=0.15 in equation 28, in order to achieve an equal amount of noise due to the random spiking as in the case of behavioural activity (see Supplementary Figure S2).”

Reviewer 2:I think the authors of this article need to be clear about the shortcomings of RL. They should devote some space in the discussion to noting neuroscience data that has not been addressed yet. They could note that most components of their RL framework are still implemented as algorithms rather than neural models. They could note that most RL models usually don't have neurons of any kind in them and that their own model only uses neurons to represent state and successor representations, without representing actions or action selection processes. They could note that the agents in most RL models commonly learn about barriers by needing to bang into the barrier in every location, rather than learning to look at it from a distance. The ultimate goal of research such as this should be to link cellular level neurophysiological data to experimental data on behavior. To the extent possible, they should focus on how they link neurophysiological data at the cellular level to spatial behavior and the unit responses of place cells in behaving animals, rather than basing the validity of their work on the assumption that the successor representation is correct.

We thank the reviewer for the important feedback. We have addressed the reviewer 2's concerns in the "Public Evaluation" section, which includes their detailed comments. In short, here, we made substantial changes to refer more thoroughly to the experimental literature, discuss the limitations of RL in general and the limitations of our proposed framework, discuss the link between neuronal data and behaviour, and finally, we made sure not to overstate the validity of the successor representation.

Reviewer 3:1. Could the authors elaborate more on the connection between the biological replays that are observed in a different context in the brain and the replays implemented in their model? Within the modeling context, when are replays induced upon learning in a novel environment, and what is the influence of replays when/if they are generated upon revisiting the previously seen/navigated environment?

We thank the reviewer for the interesting questions. Within our modelling context, we speculate that replays may contribute to the learning of the SR. To understand why this may be beneficial, we show that in our framework, learning is faster in a novel environment when the proportion of replays is larger. Replays, because they rely on the MC algorithm, are great for learning quickly as they are fast to override obsolete weights. Specifically for Figure 4, we implemented a probability for replays that decays exponentially with the number of epochs in a novel environment, but other schemes that introduce more replays in novel environments than in familiar ones should lead to similar conclusions. We also show that, when replays are generated in familiar environments, they still contribute to learning the same SR but introduce more variance. We also argue that replays can be used to imagine novel trajectories (similar to ideas in model-based planning) and thus update the SR without actually walking the trajectory (Figure 5). In summary, we believe that replays can functionally serve a variety of purposes, and our framework merely proposes additional beneficial properties without claiming to explain all observed replays. For example, our framework does not encompass reverse replays. We have updated the ‘Replays’ paragraph of the Discussion section to reflect this.

“We have also proposed a role for replays in learning the SR, in line with experimental findings and RL theories Russek 2017 Momennejad 2017. In general, replays are thought to serve different functions, spanning from consolidation to planning Roscow 2021. Here, we have shown that when the replayed trajectories are similar to the ones observed during behaviour, they play the role of speeding up and consolidating learning by regulating the biasvariance trade-off, which is especially useful in novel environments. On the other hand, if the replayed trajectories differ from the ones experienced during wakefulness, replays can play a role in reshaping the representation of space, which would suggest their involvement in planning. Experimentally it has been observed that replays often start and end from relevant locations in the environment, like reward sites, decision points, obstacles or the current position of the animal FreyjaOlafsdottir 2015, Pfeiffer 2013, Jackson 2006,Mattar 2017. Since these are salient locations, it is in line with our proposition that replays can be used to maintain a convenient representation of the environment. It is worth noticing that replays can serve a variety of functions, and our framework merely proposes additional beneficial properties without claiming to explain all observed replays. For example, next to forward replays, also reverse replays are ubiquitous in hippocampus Pfeiffer 2020. The reverse replays are not

included in our framework, and it is not clear yet whether they play different roles, with some evidence suggesting that reverse replays are more closely tied to the reward encoding Ambrose 2016.”

2. The model is composed of CA1 and CA3, what are the roles of the other hippocampal subregions in learning predictive maps? From the reported results, it looks like it may be possible that prediction-based learning can be successfully achieved simply via the CA1-CA3 circuit. Are there studies (e.g., lesioned) that show this causal relationship to behavior? Along this line, what are the potential limitations of the proposed framework in understanding the circuit computation adopted by the hippocampus?

We thank the reviewer for the feedback. We have added the following paragraph in the discussion to address this point:

“There are three different neural activities in our proposed framework: the presynaptic layer (CA3), the postsynaptic layer (CA1), and the external inputs. These external inputs could for example be location-dependent currents from the entorhinal cortex, with timings guided by the theta oscillations. The dependence of CA1 place fields on CA3 and entorhinal input is in line with lesion studies (see e.g. Brun2008, Hales2014, Oreilly2014). It would be interesting for future studies to further dissect the role various areas play in learning cognitive maps.”

3. Do the authors believe that the plasticity rules/computational principles observed within the 2-layer model are specific to the CA1-CA3 circuit? Can these rules be potentially employed elsewhere within the medial temporal lobe or sensory areas? What are the model parameters used that could suggest that the observed results are specific to hippocampus-based predictive learning?

We thank the reviewer for this important question. We refer to the following paragraph in the discussion regarding this topic:

“Notably, even though we have focused on the hippocampus in our work, the SR does not require predictive information to come from higher-level feedback inputs. This framework could therefore be useful even in sensory areas: certain stimuli are usually followed by other stimuli, essentially creating a sequence of states whose temporal structure can be encoded in the network using our framework. Interestingly, replays have been observed in other brain areas besides the hippocampus Kurth-Nelson2016, Staresina2013. Furthermore, temporal difference learning in itself has been proposed in the past as a way to implement prospective coding Brea2016”

4. The analytical illustration linking the proposed model with reinforcement learning is well executed. However, in practice, the actual implementation of reinforcement learning within the model is unclear. Given the sample task provided where animals are navigating a simple environment, how can one make use of value-based learning to enhance behavior? Explicit discussion on the extent to which reinforcement learning is related to the actual computation potentially needed to navigate sensory environments (both learned and novel) would be really helpful in understanding the link between the model to reinforcement learning.

We thank the reviewer for this important remark. We have added Figure 5—figure supplement S2 and updated the discussion to address this point. Since we modelled our neurons to integrate the synaptic EPSPs and generated spikes using an inhomogeneous Poisson process based on the depolarization, the firing rate is proportional to the total synaptic weights. Therefore, the successor representation can be read out simply by a downstream neuron. Moreover, since the value of a state is defined by the inner product between the successor matrix and the reward vector, it is sufficient for the synaptic weights to the downstream neuron to learn the reward vector, and the downstream neuron will then encode the state value in its firing rate. We performed additional simulations and have added a supplementary figure (Figure 5—figure supplement S2) showing this setup.

Besides the supplementary figure (Figure 5—figure supplement S2), we added the following paragraph in the Discussion section:

“Since we modelled our neurons to integrate the synaptic EPSPs and generate spikes using an inhomogeneous Poisson process based on the depolarization, the firing rate is proportional to the total synaptic weights. Therefore, the successor representation can be read out simply by a downstream neuron. Moreover, since the value of a state is defined by the inner product between the successor matrix and the reward vector, it is sufficient for the synaptic weights to the downstream neuron to learn the reward vector, and the downstream neuron will then encode the state value in its firing rate (see Figure 5—figure supplement S2). While the neuron model used is simple, it will be interesting for future work to study analogous models with nonlinear neurons.”

5. Subplots both within and across figures seem to be of very different text formatting and sizing (such as panel F in Figure 4 and Figure 5). Please reformat them accordingly.

Thank you, we have reformatted the figures.

Reviewer #2 (Recommendations for the authors):Important: Note that the page numbers refer to the page in the PDF, which is their own page number-1 (due to eLife adding a header page).Page 3 – "smoothly… and anything in between" – this is overstated and should be removed.

We thank the reviewer for the feedback. We have adapted the sentence as follows:

“We show mathematically that our proposed framework smoothly connects a temporally precise spiking code akin to replay activity with a rate based code akin to behavioural spiking.”

While we agree that our model implements only one type of temporal code, it is important to stress that there is a smooth transition between this temporal code and a pure rate encoding of the state. The larger the time T in each state, the less precise the spikes become organised in time. In all cases, the learning dynamics have the same fixed point, namely, they converge to the successor representation. We believe that this fact is non-trivial. Furthermore, we show how this smooth transition changes how the fixed point (SR) is reached. On one extreme (T=0), algorithmically it uses TD(1), on the other extreme (T=infinity) TD(0) and all intermediate cases implement a value of λ between 0 and 1.

Page 3 – "don't need to discretize time…". Here and elsewhere there should be citations to the work of Doya, NeurIPS 1995, Neural Comp 2000 on the modeling of continuous time in RL.

Thank you, we have added those citations

Page 3 – "using replays" – It is very narrowminded to assume that all replay do is set up successor representations. They could also be involved in model-based planning of behavior as suggested in the work of Johnson and Redish, 2012; Pfeiffer and Foster, 2018; Kay et al. 2021 and modeled in Hasselmo and Eichenbaum, 2005; Erdem and Hasselmo, 2012 and Fenton and Kubie, 2012.

We thank the reviewer for the feedback, but assure that we did not mean to assume that is all replays do. In fact, we are well aware of other potential benefits of replays in memory consolidation, model-based planning etc. We merely propose to add another potential benefit of replays to the existing hypotheses, namely that they can be used to learn and in doing so they can reduce bias and learn offline. We have adapted the ‘Replays’ paragraph in the discussion to reflect this and have also adapted the highlighted text to make this clearer:

“Finally, replays have long been speculated to be involved in learning models of the environment (Eichenbaum, 2005; Hasselmo and Erdem and Hasselmo, 2012 and Fenton and Kubie, 2012; Johnson and Redish, 2012; Pfeiffer and Foster, 2018; Kay et al. 2021;). Here, we investigate how replays could play an additional role in learning the SR cognitive map”

Page 3 – They assume that STDP can occur during replay, but evidence for STDP during replay is unclear. McNaughton's lab showed that LTP is less likely to be induced during the modulatory states during which sharp-wave ripple replay events occur. They should look for citations that have actually shown LTP induction during the replay state.

Thank you for this comment. We have added the following sentence in the "Replay" paragraph of the discussion.

“Moreover, while indirect evidence supports the idea that replays can play a role during learning Igata2020, it is not yet clear how synaptic plasticity is manifested during replays Fuchsberger2022.”

Page 4 – Marr's three levels – They should remove this discussion about Marr's three levels as I think the implementation level is relatively sparse and the behavioral level is also relatively sparse.

We believe that this link between computation, algorithm and implementation is actually a strength of the proposed work. Typically, only one or at most two levels are discussed, without a more holistic view. We believe that our work (and e.g. George et al.) goes one step beyond by making the link between the implementation (spiking neurons with STDP), algorithm (TD learning) and computational theory (SR / predictive cognitive maps) explicit. While of course, these modelling studies are abstractions of reality, we believe this is not trivial and would like to maintain this paragraph to stimulate future research to bridge these levels as well.

Page 4 – "The hippocampus has long been thought" – It's astounding that the introduction only cites two experimental papers (O'Keefe and Dostrovsky, 1971; Mehta et al. 2000) and then the start of the Results section makes a statement like this and only cites Stachenfeld et al. 2017 as if it were an experimental paper. There are numerous original research papers that should be cited for the role of hippocampus in behavior. They should cite at least five or six so that the reader doesn't get the impression all o this work started with the holy paper by Stachenfeld et al. 2017. For example, they could again cite O'Keefe and Nadel, 1978 for the very comprehensive review of the literature up to that time, plus the seminal work of Morris et al. 1982 in Morris water maze and Olton, 1979 in 8-arm radial maze and perhaps some of the work by Aggleton and by Eichenbaum on spatial alternation.

We agree and thank the reviewer for the suggestion. We have now expanded the citations of relevant experimental work.

Page 5 – The description of successor representations is very one dimensional. They should mention how it can be expanded to two dimensions.

To address this, we perform a new simulation in a 2D environment (Supplementary Figure 2) as well as a discussion on the generality of the approach to any dimension in section 2.1

“Even though we introduced the linear track as an illustrative example, the SR can be learned in any environment (see Figure 1—figure supplement S1 for an example in an open field)”.

Page 5 – "Usually attributed to model based…". They cannot just talk about SR being model free. Since this section was supposed to be for neuroscientists, they need to clearly explain the distinction between model free and model-based RL, and describe why successor representations are not just model-based RL, but instead provide a look-up table of predictive state that does NOT involve model-based planning of behavior. The blog of Vitay gives a much better overview that compares model-free, model-based and successor representations:https://julien-vitay.net/post/successor_representations/ – This needs more than just a citation – there should be a clear description of model-based and model-free RL in contrast to SR, and Vitay is an example of that.

We have now extended this paragraph with a more thorough explanation of model-free and model-based RL, including an example of what happens when a reward location changes in all three cases.

Because of this predictive information, the SR allows sample-efficient re-learning when the reward location is changed Gershman2018. In reinforcement learning, we tend to distinguish between model-free and model-based algorithms. The SR is believed to sit inbetween these two modalities. In model-free reinforcement learning, the aim is to directly learn the value of each state in the environment. Since there is no model of the environment at all, if the location of a reward is changed, the agent will have to first unlearn the previous reward location by visiting it enough times, and then is able to re-learn the new location. In modelbased reinforcement learning, a precise model of the environment is learned, specifically, single-step transition probabilities between all states of the environment. Model-based learning is computationally expensive, but allows a certain flexibility. If the reward changes location it is immediate to derive the updated values of the states. As we have seen, however, the SR can re-learn a new reward location somewhat efficiently, although less so than model-based learning. The SR can also be efficiently learned using model-free methods and allows us to easily compute values for each state, which in turn can guide the policy Dayan1993, Russek2017, Momennejad2017. This position between model-based and model-free methods makes the SR framework very powerful, and its similarities with hippocampal neuronal dynamics have led to increased attention from the neuroscience community. Finally, in our examples above, we considered an environment made up of a discrete number of states. This framework can be generalised to a continuous environment represented by a discrete number of place cells.

Page 5 – Related to this issue – they need to repeatedly address the fact that Successor representations are just an hypothesis contrast with model-based behavior, and repeatedly throughout the paper discuss that model-based behavior could still be the correct accounting for all of the data that they address.

We thank the reviewer for pointing out that the successor representation is only one of the possible hypotheses.

We have proceeded to address this for the Frank and Cheng data (page 14, ‘Please note, however, that other mechanisms besides the Successor Representation could account for these results, including model-based reinforcement learning.), and in the final section of the shock experiment (page 16, ‘It is important to note here that, while we are suggesting a potential role for the SR in solving this task, the data itself would also be compatible with a model-based strategy. In fact, experimental evidence suggests that humans may use a mixed strategy involving both model-based reinforcement learning and the successor representation [Momennejad et al., 2017].’).

Page 5 – "similar to (Mehta et al. 2000)" – Learning in the CA3-CA1 network has been modeled like this in many previous models that should be cited here including McNaughton and Morris, 1987; Hasselmo and Schnell, 1994; Treves and Rolls, 1994; Mehta et al. 2000; Hasselmo, Bodelon and Wyble, 2002.

Thank you, we have added the citations

Page 6 – Figure 1d looks like the net outcome of the learning rule in this example is long-term depression. Is that intended? Given the time interval between pre and post, it looks like it ought to be potentiation in the example.

This depends on the pre-post spike timing, and in this example the postsynaptic spike is too far from the presynaptic spike, leading to depression. When moving the postsynaptic spike closer to the presynaptic spike, potentiation would occur. The plasticity rule qualitatively results in three regions depending on the spike timing: depression (post-pre), potentiation (pre-post with small interval), and depression (pre-post with large interval). Qualitatively it is in line with e.g. Shouval et al. 2002

Page 7 – They should address the problem of previously existing weights in the CA3 to CA1 connections. For example, what if there are pre-existing weights that are strong enough to cause post-synaptic spiking in CA1 independent of any entorhinal input? How do they avoid the strengthening of such connections? (i.e. the problem of prior weights driving undesired learning on CA3-CA1 synapses is addressed in Hasselmo and Schnell, 1994; Hasselmo, Bodelon and Wyble, 2002, which should be cited).

This is an important point, and since our model guarantees a stable fixed point of the learning dynamics, the initial conditions are not influencing the final convergence. To show this explicitly, we have performed a new simulation and added Figure 1—figure supplement S2 to illustrate this.

This is reflected in the manuscript:

“As a proof of principle, we show that it is possible to learn the SR for any initial weights (Figure 1—figure supplement S2), independently of any previous learning in the CA3 to CA1 connections.”

Page 7 – "Elegantly combines rate and temporal" – This is overstated. The possible temporal codes in the hippocampus include many possible representations beyond just one step prediction. They need to specify that this combines one type of possible temporal code. I also recommend removing the term "elegant" from the paper. Let someone else call your work elegant.

We thank the reviewer for the feedback. We have modified the manuscript to reflect the comments:

“As we will discuss below, this framework, therefore, combines learning based on rate coding as well as temporal coding.”

Page 7 – "replays for learning" – as noted above in experiments LTP has not been shown to be induced during the time periods of replay – sharp-wave ripple replay events seem to be associated with lower cholinergic tone (Buzsaki et al. 1983; VandeCasteele et al. 2014) whereas LTP is stronger when Ach levels are higher (Patil et al. J. Neurophysiol. 1998). This is not an all-or-none difference, but it should be addressed.

Thank you for this comment. We have added the following sentence in the "Replay" paragraph of the discussion.

“Moreover, while indirect evidence supports the idea that replays can play a role during learning Igata2020, it is not yet clear how synaptic plasticity is manifested during replays Fuchsberger2022.”

Page 7 – "equivalent to TD…". Should this say "equivalent to TD(0)"?

We changed the phrasing in the manuscript to: “equivalent to TD(λ)”.

Page 8 – "Bootstrapping means that a function is updated using current estimates of the same function…". This is a confusing and vague description of bootstrapping. They should try to give a clearer definition for neuroscientists (or reduce their reference to this).

We have now changed the explanation in the manuscript in the corresponding section.

From a reinforcement learning perspective, the TD(0) algorithm relies on a property called bootstrapping. This means that the successor representation is learned by first taking an initial estimate of the SR matrix (i.e. the previously learned weights), and then gradually adjusting this estimate (i.e. the synaptic weights) by comparing it to the states in the environment the animal actually visits. This comparison is achieved by calculating a prediction error, similar to the widely studied one for dopamine neurons Schultz1997. Since the synaptic connections carry information about the expected trajectories, in this case, the prediction error is computed between the predicted and observed trajectories (see Methods).

The main point of bootstrapping, therefore, is that learning happens by adjusting our current predictions (e.g. synaptic weights) to match the observed current state. This information is available at each time step and thus it allows learning over long timescales using synaptic plasticity alone. If the animal moves to a state in the environment that the current weights deem unlikely, potentiation will prevail and the weight from the previous to the current state will increase. Otherwise, the opposite will happen. It is important to notice that the prediction error in our model is not encoded by a separate mechanism in the way that dopamine is thought to do for reward predictionSchultz1997. Instead, the prediction error is represented locally, at the level of the synapse, through the depression and potentiation terms of our STDP rule, and the current weight encodes the current estimate of the SR (see Methods). Notably, the prediction error updates result in a total update equivalent to the TD(\λ) update. This mathematical equivalence ensures that the weights of our neural network track the TD(\λ) update at each state, and thus stability and convergence to the theoretical values of the SR. We therefore do not need an external vector to carry prediction error signals as proposed in Gardner2018, Gershman2018. In fact, the synaptic potentiation in our model updates a row of the SR, while the synaptic depression updates a column.

Page 9 – Figure 2 – Do TD λ and TD zero really give equivalent weight outputs?

The theoretical value of the SR is the same, independently of the algorithm used to learn it. TD λ, for any value of λ, converges to this theoretical value. No tuning is needed for this, it is mathematically guaranteed that the algorithm converges. Thus, also for λ=0 the results are exact.

Page 8 – "that are behaviorally far apart" – I don't understand how this occurs.

See explanation above.

page 10 – "dependency of synaptic weights on each other as discussed above." This was not made sufficiently clear either here or above.

See explanation above.

Page 10 – "dependency of synaptic weights on each other" This also suggests a problem of stability if the weights can start to drive their own learning and cause instability – how is this prevented?

See explanation above. The dynamics have a fixed point of convergence (the SR). No matter what the initial weights are, it is guaranteed to converge to the SR. We illustrated that in the new Supplementary Figure 7.

Page 10 – "average of the discounted state occupancies" – this would be uniform without discounting but what is the biological mechanism for the discounting that is used here?

This is an important point. No extra mechanism is needed to induce the discount in our framework. Instead, the decaying LTP window is sufficient to drive the discounting of states further away.

Page 10 – "due to the bootstrapping" – again this is unclear – can be improved by giving a better definition of bootstrapping and possibly by referring to specific equation numbers.

We have now changed the explanation in the manuscript in the corresponding section, see comment above.

Page 11 – "exponential dependence" what is the neural mechanism for this?

In our case, it is the LTP window which exponentially decays.

Page 11 – "Ainsley" is not a real citation in the bibliography. Should fix and also provide a clearer definition (or equation) for hyperbolic.Page 11 – "elegantly combines two types of discounting" – how is useful? Also, let other people call your work elegant.

Very true! We have now rephrased our statement. Hyperbolic discounting has been observed in neuroeconomics experiments, whereas reinforcement learning algorithms normally rely on exponential discounting for its favourable mathematical properties. This has been however considered confusing when applying reinforcement learning to behaviour. Our model uses exponential discounting at a neural level and in space, but can account for the findings with hyperbolic discounting.

Page 11 – how does discounting depend on both firing rate and STDP -- should provide some explanation or at least refer to where this is shown in the equations.

We have now added a reference to the equation.

page 13 – "Cheng and Frank" – this is a good citation, but they could add more here on timing of replay events.

We added the following sentence regarding the replay timing:

“Please note that, while we implemented an exponentially decaying probability for replays after entering a novel environment, different schemes for replay activity could be investigated.”

Page 15 – This whole section on the shock experiment starts with the assumption of a successor representation. As noted above, they need to explicitly discuss the important alternate hypothesis that the neural activity reflects model-based planning that guides the behavior in the task (and could perhaps better account for the peak of occupancy at the border of light and dark).

We have changed the text to reflect this suggestion.

Page 16 – "mental imagination" – rather than using it for modifying SR, why couldn't mental imagination just be used for model-based behavior?

We believe it is indeed possible that mental imagination can be used for model-based behaviour. However, it is possible that humans use a mixed strategy involving both modelbased learning and the successor representation, as suggested by the findings in Momennejad et al. 2017.

Page 17 – "spiking" – again, if they are going to refer to their model as a "spiking" model, they need to add some plots showing spiking activity.

One example of the spiking activity of our model can be found in Figure 2 panel A.

Reviewer #3 (Recommendations for the authors):

We have responded to the reviewer 3's concerns at the beginning of the rebuttal, in the "Essential Revision" section.

[Editors' note: further revisions were suggested prior to acceptance, as described below.]

Reviewer #1 (Recommendations for the authors):The revised article has only partly resolved my confusion.My main issue was the following: in the proposed feed-forward model, the synaptic weights between the two layers learn the entries of the successor matrix. If external inputs were fed only to the first layer, the second layer would directly read out the successor representation (this is suggested in Figure 1 E-F, but not explicitly mentioned in the text as far as I can tell). Instead, in the model, both layers are driven by external inputs representing the current state. This is crucial for learning, but it implies that the activity of the second layer mixes the successor representation with a representation of the current state itself. Learning and readout, therefore, seem antagonistic. It would be worth explaining this fact in the main text.

We thank the reviewer for the question, and we now added an extra explanation in the main text. In brief, the CA1 layer indeed mixes the SR with a representation of the current state. However, we argue that learning and readout are not necessarily antagonistic. We suggest here a few possibilities to resolve this conflict:

1. Since the external current in CA1 is present for only a fraction of the time T in each state, the readout might happen during the period of CA3 activation exclusively.

2. The readout may be over the whole time T but becomes noisier towards the end. It is worth noting that, even in the case where the readout is noisy, the distortion would be limited to the diagonal elements of the matrix (see equation 13 and 15 in the Methods section, and Figure 5 – supplement 2, panels B and C).

3. Learning and readout may be separate mechanisms, where during readout only the CA3 driving current is present. This could be for instance signaled by neuromodulation (e.g. acetylcholine has been associated to attention and arousal during spatial learning [1,2,3,4,5]), or it could be that readout happens during replays with a population of neurons.

4. The weights to or activation functions of the readout neuron may learn to compensate for the distorted signal in CA1.

We have now included this explanation in the main text in the Discussion, in the Reinforcement Learning subchapter.

In their reply, the authors clarify that the external inputs drive the activity of the second layer only for a limited time (20%). As far as I can tell, in the text, this is mentioned explicitly only in the legend of Figure 5-S2. That seems to imply that there is a large difference in the duration of the external inputs to the two layers. How can that be justified?More importantly, it seems that varying the value of the delay should lead to a tradeoff between the accuracy of learning and the accuracy of the subsequent readout in the second layer. Is that the case? It would be useful to have a figure where the delay is varied.

As suggested, we added two figures (Figure 5—figure supplement 3, Figure 5—figure supplement 4) where we varied this delay. In our model, the external inputs are first active in CA3 and afterwards in CA1 (equation 10 and 11). We model these inputs as step currents. We call the overall time in each state T, and say that the external current for CA3 has duration θ, while the external current for CA1 has duration ω omega. If we assume that there is no pause in between these two stimuli, we have ω = T – θ. We can therefore define the delay between the two stimuli as: θ/T.

As the reviewer points out, the delay value θ/T that we chose for Figure 5 – Supplement 2 was 0.8. However, this was an arbitrary choice and other values are possible. We explore here what happens when we vary the delay parameter θ/T:

– Impact on the reinforcement learning representation

The delay parameter θ/T influences the reinforcement learning representation, as it can be seen from equations 22 and 23. When the delay is longer, the value for γ increases and λ decreases (Figure 5 —figure supplement 3, panels b-c). However, it is important to notice that γ and λ are determined by other biological values as well (see equations 22 and 23), such as the depression amplitude (Figure 5 —figure supplement 3, panels e-f) or the decay of the potentiation learning window. The value we choose for the delay is therefore flexible: even if we want to learn the SR with certain fixed γ or λ, we can always vary other parameters in the spiking model.

– Impact on learning and readout

The strength of the place-tuned input to CA1 also depends on the choice of the delay (see equation 18). Typically, the CA1 place-tuned input strength increases with the delay, except for long values of T or low depression amplitudes, where the input remains fairly constant (Figure 5 —figure supplement 3, panels a and d respectively). Therefore, the longer the CA1 place-tuned input lasts, the weaker it has to be to ensure the proper learning. Intuitively, this means that we can compensate for a shorter duration of the external current by increasing its strength. The distortion of the readout thus remains more or less constant independently of our choice of delay (Figure 5 —figure supplement 4). We therefore do not find a trade-off between accuracy of learning and accuracy of readout, and the distortion in the value could be corrected for example by one of the mechanisms proposed above.

We addressed the above points in the Discussion, in the Reinforcement Learning subchapter.

References:

Micheau, J. and Marighetto, A. Acetylcholine and memory: a long, complex and chaotic but still living relationship. Behav. Brain Res. 221, 424–429 (2011)

Hasselmo, M. E. and Sarter, M. Modes and models of forebrain cholinergic neuromodulation of cognition. Neuropsychopharmacology 36, 52–73 (2011).

Robbins, T. W. Arousal systems and attentional processes. Biol. Psychol. 45, 57–71 (1997).

Teles-Grilo Ruivo, L. M. and Mellor, J. R. Cholinergic modulation of hippocampal network function. Front. Synaptic Neurosci. 5, 2 (2013).

Palacios-Filardo, Jon, et al. "Acetylcholine prioritises direct synaptic inputs from entorhinal cortex to CA1 by differential modulation of feedforward inhibitory circuits." Nature communications 12.1 (2021): 1-16.